# Spend Wisely: Maximizing Post-Training Gains in Iterative Synthetic Data Bootstrapping

**Pu Yang**[*]
Peking University

**Yunzhen Feng**[*]
New York University

**Ziyuan Chen**[*]
Peking University

**Yuhang Wu**
UC Berkeley

**Zhuoyuan Li**[†]
National University of Singapore

## Abstract

Modern foundation models often undergo iterative "bootstrapping" in their post-training phase: a model generates synthetic data, an external verifier filters out low-quality samples, and the high-quality subset is used for further fine-tuning. Over multiple iterations, the model performance improves, raising a crucial question: How should the total budget for generation and training be allocated across iterations to maximize final performance? In this work, we develop a theoretical framework for analyzing budget allocation strategies. Specifically, we show that constant policies fail to converge with high probability, while increasing policies—particularly exponential growth policies—exhibit significant theoretical advantages. Experiments on image denoising with diffusion probabilistic models and math reasoning with large language models show that both exponential and polynomial growth policies consistently outperform constant policies, with exponential policies often providing more stable performance.

## 1 Introduction

Supervised fine-tuning is a critical stage in training large language models (LLM). Pre-training supplies them with broad linguistic and factual knowledge, but task-specific skills—such as tool use (Schick et al., 2023), reasoning pattern (Gandhi et al., 2025), and agentic behavior (Shao et al., 2023)—emerge when the model is further refined on carefully curated, supervised examples. Securing these high-quality human-annotated data, however, remains a major bottleneck, as it requires domain expertise and substantial resources.

To address this limitation, synthetic data have emerged as a promising alternative that offers scalability and cost-effectiveness. Despite concerns about potential risks of model collapse (Shumailov et al., 2024; Dohmatob et al., 2024b), synthetic data is typically selected and verified before use in post-training (Feng et al., 2024; Setlur et al., 2024), ensuring its quality. A common paradigm for leveraging synthetic data employs an iterative bootstrapping process (Zelikman et al., 2022; Trung et al., 2024): the model generates synthetic data, rewards or verifiers are used to filter and select high-quality data, and the model is then fine-tuned on selected data. This process is repeated iteratively to fully improve performance. An illustration of this approach is provided in Figure 1.

However, for practitioners who implement this approach, an important question arises: How should a fixed computational budget be allocated to decide the amount of synthetic data to generate and select at each iteration to maximize performance?

---

[*]Equal contribution.

[†]Corresponding Author. `zy.li@nus.edu.sg`

39th Conference on Neural Information Processing Systems (NeurIPS 2025).

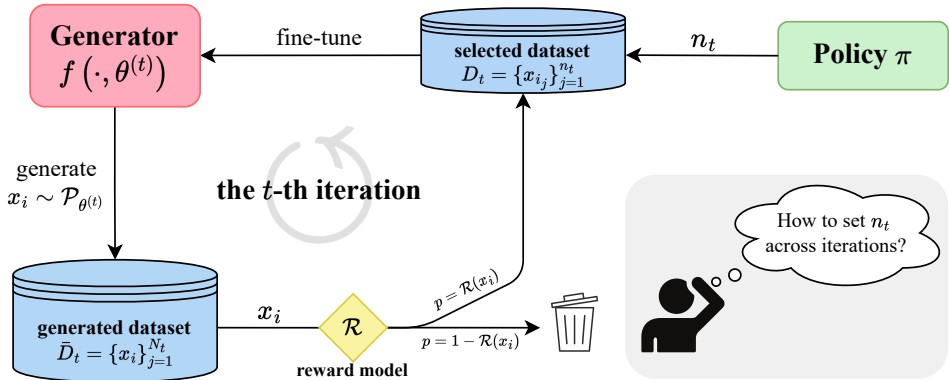

Figure 1: **Iterative learning with synthetic data.** In this framework, synthetic data is generated, filtered using a reward model, and the selected data is used to further train the generator. The budget policy is defined as the quantity of data retained after selection, $n_t$. Our goal is to identify the optimal policy across iterations to achieve the best final performance, given a fixed budget.

In this paper, we establish some foundational principles for crafting optimal strategies for synthetic data generation across iterations. To the best of our knowledge, this is the first attempt to address this problem. We begin with a theoretical analysis of policies that control the amount of selected synthetic data used in each iteration, as illustrated in Figure 1. In a simplified setting with Gaussian data and exponential reward functions, we could identify the optimal policy. Contrary to common strategies that maintain a fixed number of synthetic data across iterations (constant policies), our analysis shows that the optimal strategy requires exponentially increasing the amount of data at each iteration.

Furthermore, in more general settings with only mild assumptions about regularity, we demonstrate that constant policies fail to converge with high probability. In contrast, policies that increase the amount of synthetic data across iterations can converge to optimal performance. Among such increasing policies, we prove that exponential growth policies guarantee exponential convergence, and in the worst-case scenarios, they achieve no larger computational costs compared to polynomial growth policies when attaining similar near-optimal performance.

Building on these theoretical insights, we validate our findings with two experiments: an image-denoising task using diffusion probabilistic models (DPMs), and a math-reasoning task with large language models (LLMs). Across these experiments, exponential and linear (polynomial) growth policies outperform constant policies. Moreover, the best exponential policies match or exceed the performance of linear policies in every case, confirming their theoretical robustness.

We summarize our contribution as follows.

- We formulate the problem of optimizing post-training gains under a limited budget for iterative learning in Section 3.1.
- In a solvable Gaussian setting, we prove the optimality of an exponential growth policy in Section 3.2 and validate it through numerical simulations.
- In a more general setting, we demonstrate in Section 4 that constant policies fail to converge to the optimal reward with high probability, while increasing policies ensure convergence. Among these, the exponential growth policy achieves an exponential convergence rate and outperforms polynomial growth policies in the worst case.
- Experiments on image denoising (diffusion probabilistic model) and math reasoning (large language model) in Section 5 confirm that exponential and polynomial growth policies outperform constant policies, with exponential policies exhibiting greater stability.

## 2   Related Work

**Synthetic Data and Iterative Bootstrapping.** The iterative bootstrapping framework, illustrated in Figure 1, is widely used for training foundational models in both text and vision domains. In this approach, synthetic data are generated and then iteratively refined or filtered to produce new

training sets that more closely align with the target objectives. For large language models (LLMs) and vision language models (VLMs), this approach has been applied to instruction following (Touvron et al., 2023) and alignment tasks (Dong et al., 2023), and vision question answering (Yang & Dong, 2025), wherein reward models filter synthetic data to retain only high-quality samples. Recently, it has been particularly effective in improving reasoning abilities, as first demonstrated in Zelikman et al. (2022). Subsequent works (Lehnert et al., 2024; Su et al., 2024; Trung et al., 2024; Guan et al., 2025; Singh et al., 2025) have extended this approach to generate diverse reasoning paths with search algorithms. Beyond text, iterative bootstrapping has also been utilized to generate training data for image segmentation (Kirillov et al., 2023), create image-text pairs (Fan et al., 2023, 2024), and synthesize images to address distribution shifts (Hemmat et al., 2024; Azizi et al., 2023). However, in these empirical works, a principled method for controlling generation in each iteration is still missing.

To our knowledge, Ferbach et al. (2024) provides the only theoretical analysis of this paradigm, proving convergence to the optimal policy under the infinite-sample assumption. Crucially, their framework fails to characterize finite sample requirements, offering little actionable guidance to practitioners implementing this framework. We address this gap by establishing optimal computational budget allocation strategies across iterations to achieve maximum performance gains.

**Model Collapse**. A series of papers has shown that the incorporation of synthetic data into the training corpus can lead to performance degradation, either in a single training instance or iteratively over time (Shumailov et al., 2023; Alemohammad et al., 2024; Dohmatob et al., 2024b,a, 2025). Feng et al. (2024) has explored the use of data selection to overcome model collapse by employing synthetic data in a single generation, combined with verification. In our work, we also leverage verification, but focus on addressing the question of how to optimally generate synthetic data during learning.

## 3    Problem Formulation and Preliminary Analysis

In this section, we formalize the iterative learning setting in Section 3.1 and then analyze optimal budget allocation strategies with a simple Gaussian case in Section 3.2.

### 3.1    Problem Formulation

The setup is illustrated in Figure 1. Our goal is to train a parameterized generative model $f(\cdot; \theta)$, for example, given by a DPM or an LLM, that generates data $x \sim \mathcal{P}_\theta$. A reward model $\mathcal{R}$ evaluates the quality of the generated data, serving as a measure of the model performance. We aim to train the generative model to maximize the expected reward

$$r(\theta) = \mathbb{E}_{x \sim \mathcal{P}_\theta}[\mathcal{R}(x)],$$

where we assume $\mathcal{R}(x) \in [0, 1]$.

The iterative training process leverages the generative nature of the model to perform self-generation and refinement based on reward signals. Specifically, we start with an initial model $\theta^{(0)}$. At the $(t + 1)$-th iteration, the model undergoes improvement through the following three steps:

- **Generation step**: Generate $N_t$ data by $f(\cdot; \theta^{(t)})$.
- **Selection step**: For each synthetic data point $x$, include it into dataset $D_t$ with probability $\mathcal{R}(x)$. Suppose $n_t$ of the $N_t$ data are selected to form the dataset $D_t$, formally, $|D_t| = n_t$.
- **Updating step**: Update $\theta^{(t)}$ to $\theta^{(t+1)}$ with the dataset $D_t$.

This iterative approach generates additional training data using synthetic samples, with the reward-guided selection process ensuring its quality. The selected samples are then used to further enhance the model's generative performance (Feng et al., 2024). This process, carried out iteratively to maximize performance gains, is particularly effective in scenarios where high-quality specialized data is scarce, for example, in mathematical reasoning (Zelikman et al., 2022; Guan et al., 2025; Singh et al., 2025). The algorithm framework is formalized in Algorithm 1. In the selection step, the data may be selected with noise. We present a simple form here for ease of understanding, while our results extend to the noisy case in Appendix B.5.

When practitioners use this algorithm, a key challenge is determining how to set $N_t$ and $n_t$ across iterations. This decision is particularly important, as the generation process of foundation models unually involves multiple forward propagations, and consequently, the generation cost often exceeds the training cost. Given a limited budget, it is crucial to design the process in a way that maximizes post-training gains.

Specifically, we focus on identifying the optimal policy of $N_t$ and $n_t$ correlated with the generation cost and the training cost, respectively. Given the reward model and generator, $N_t$ can be viewed as a random variable dependent on $n_t$ and the selection rate. Therefore, we define the policy over $n_t$ as $\pi : \{n_t\}_{t=0,1,\cdots}$, and our objective is to identify the best scheme for $\pi$.

---

**Algorithm 1** Iterative learning with synthetic data

---

1: **Input:** An initial generative model $f(\cdot; \theta^{(0)})$, a reward model $\mathcal{R}$, and a policy $\pi : \{n_t\}_t$ with a termination time $T$.
2: **Output:** A fine-tuned generative model $f(\cdot; \theta^{(T)})$.

3: **for** $t \leftarrow 0$ to $T - 1$ **do**
4:     Initialize selected synthetic dataset $D_t = \varnothing$.
5:     **while** $|D_t| < n_t$ **do**
6:         Generate a sample $x \sim \mathcal{P}_{\theta^{(t)}}$.
7:         Add $x$ to $D_t$ with probability $\mathcal{R}(x)$.
8:     **end while**
9:     Update $\theta^{(t)}$ to $\theta^{(t+1)}$ using the dataset $D_t$.
10: **end for**

---

## 3.2 Gaussian Case Study

To begin, we present a warm-up example with Gaussian data, where the optimal policy can be derived analytically. This example provides theoretical insight into the general iterative learning framework. The setup includes Gaussian data, an exponential reward function, and maximum likelihood estimation (MLE) as the updating algorithm.

**Gaussian Generator and Exponential Reward Model.** To simplify the setting, we consider the generative model as a one-dimensional Gaussian distribution $\mathcal{P}_\theta = \mathcal{N}(\theta, \sigma^2)$, where the mean $\theta$ is the parameter to learn, and the variance $\sigma^2$ is fixed. The reward model is an exponential function $\mathcal{R}(x) = \exp\left(-x^2/(2\kappa^2)\right)$, where $\kappa$ is also fixed.

**Updating via MLE.** In Algorithm 1, we employ MLE as the updating algorithm at line 9. Specifically, the parameter $\theta^{(t+1)}$ updated in the $t$-th iteration is obtained by

$$\theta^{(t+1)} = \arg\max_\theta \prod_{x \in D_t} p_\theta(x),$$

where $p_\theta(x)$ is the density function of the distribution $\mathcal{P}_\theta$.

**Optimization Problem.** We aim to find the optimal iterative learning policy within this setup. We assume that the cost of sampling the Gaussian distribution is negligible, while the training cost for MLE is proportional to the size of the dataset, $n_t$. The objective is to maximize the expected performance of the generator in iteration $T$, as measured by the reward. Given a fixed budget $C$, the optimal policy will be the solution to

$$\max_{\{n_t\}_{t=0}^{T-1}} \mathbb{E}_{\theta^{(T)}} \left[ r\left(\theta^{(T)}\right) \right] \quad \text{s.t.} \quad \sum_{t=0}^{T-1} n_t \le C. \tag{1}$$

For the above setup, we can theoretically identify the optimal policy in the following theorem.

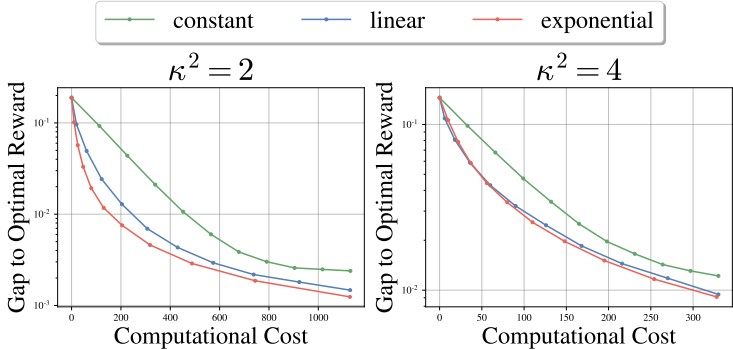

Figure 2: **Empirical results of the toy example.** We compare the exponential, constant, and linear policy, and show the gap to the optimal expected reward as a function of the computational cost ($\sum_t n_t$). All the results are averaged over 1,000 runs.

**Theorem 3.1.** *If the initial parameter satisfies $\theta^{(0)} \leq (1 + \sigma^2/\kappa^2)^T (\sigma^2 + \kappa^2)^{1/2}$, then the optimal iterative policy, i.e., the solution of the optimization problem in Equation* (1)*, is given by*

$$n_t \propto \left(1 + \frac{\sigma^2}{\kappa^2}\right)^t.$$

The proof is deferred to Appendix A and extends naturally to the high-dimensional case, as detailed in Appendix E.1. Theorem 3.1 demonstrates that the optimal policy in this setting follows an exponential scheme. The optimal strategy follows this rationale: In an iterative optimization algorithm, the contribution of an early update to the overall error decays exponentially with the total number of iterations $T$. As a result, achieving a low final error increasingly depends on the accuracy of later updates. To keep those updates precise enough, each iteration must therefore draw on an exponentially larger number of samples. We will further analyze this exponential scheme in the general analysis in Section 4.

**Experiments.** We include an empirical experiment to confirm Theorem 3.1 with data in two-dimensional space. The generator is set to $\mathcal{N}(\boldsymbol{\theta}, \boldsymbol{I}_2)$. The reward function is $\mathcal{R}(\boldsymbol{x}) = \exp\left(-\|\boldsymbol{x}\|^2/(2\kappa^2)\right)$, which favors data concentrated near the origin. $\kappa$ controls the flatness of the reward. Full details are deferred to Appendix E.2. Figure 2 plots the gap to the optimal reward as a function of the cumulative cost, defined as the total computational cost incurred up to that point. The results demonstrate that the exponential policy consistently approaches the optimal reward faster than the other schemes. While the linear policy converges slightly more slowly, the constant policy, as shown in the left figure, clearly fails to converge. These findings validate our theoretical proof. Additional results with various parameters are left in Appendix F.1.

## 4 Main Theoretical Results

We now proceed to analyze the general setting. We will show that constant policies fail to converge with high probability, whereas there exists an exponential growth policy that achieves an exponential convergence rate and outperforms polynomial alternatives in the worst case. The assumptions and setup are outlined below, while detailed proofs for the subsequent theorems are provided in Appendix B.4.

**Assumptions.** We consider a general function class for the generative model, a general loss with regularity assumptions (Assumptions B.1 and B.2), and a reward model with mild assumptions about its relationship with the loss (Assumption B.3). The formal descriptions are provided in Appendix B. The warm-up example in the previous section serves as a special case that satisfies all these assumptions. Our results also naturally extend to noisy rewards, as discussed in Appendix B.5.

**Updating via Gradient Descent.** In Algorithm 1, we use gradient descent with a learning rate $\eta > 0$ as the update algorithm at line 9. Specifically, the update for the $t$-th iteration writes

$$\theta^{(t+1)} = \theta^{(t)} - \frac{\eta}{n_t} \sum_{x \in D_t} \nabla_\theta l\left(x; \theta^{(t)}\right), \quad t = 0, 1, \cdots,$$

where $l(x; \theta)$ is the loss function. For any policy $\pi$, let $\left\{\theta_\pi^{(t)}\right\}_t$ denote the iterative trajectory of the parameter.

**Reward and Cost.** Similarly to the warm-up example, we evaluate the generative model using the expected reward $r(\theta)$. The optimal performance is the supremum of the expected reward over the whole parameter space

$$r^* = \sup_\theta r(\theta).$$

We consider a general computational cost that accounts for both generation and training, with constant coefficients $c_g$ and $c_t$ for each data point, respectively. Then, the total cost is given by

$$C(\pi, T) = \sum_{t=0}^{T-1} c_g N_t + c_t n_t.$$

The first policy class that we consider is

> • **constant policy**: $\pi_{\text{const}} : n_t = n_0, \quad t = 0, 1, \cdots$ .

This is the most standard policy, where $n_0$ can be set either as the total number of prompts in the dataset (Zelikman et al., 2022), or as a multiple of the batch size in an online setting (Guo et al., 2024). However, in the following theorem we show that constant policies fail to converge to the optimal reward with high probability.

**Theorem 4.1** (Bounded Reward for Constant policy). *Under Assumption B.3, there exists a constant $c > 0$ such that, for any $T$ and any constant policy $\pi_{\text{const}}$, with probability at least $1/4$,*

$$r^* - r\left(\theta_{\pi_{\text{const}}}^{(T)}\right) \geq cn_0^{-1/2}.$$

The theory establishes the non-convergence of constant policies, which stems from persistent random noise in the gradient. Specifically, the noise term is proportional to $n_t^{-1}$ and remains non-decaying under any constant policy, resulting in suboptimal performance.

To address this limitation, we introduce increasing policies, where $n_t$ grows monotonically over iterations. The following theorem shows that increasing policies ensure convergence to the optimal reward with high probability.

**Theorem 4.2** (Optimal Reward for Increasing Policies). *Under Assumption B.1 to Assumption B.3, for any increasing policy $\pi$ and $\varepsilon > 0$, there exists a sufficiently large $T$ such that, with probability greater than $\left(1 - \sum_{t=0}^{T-1} n_t^{-4}\right)$,*

$$r^* - r\left(\theta_\pi^{(T)}\right) \leq \varepsilon.$$

This theorem suggests that increasing policies should be used in practice to maximize performance. Hence, we consider the following two classes, namely,

> • **polynomial growth policy**: $\pi_{\text{poly}} : n_t = n_0(1 + t)^\alpha, \quad \alpha > 0, t = 0, 1, \cdots$ ;
> • **exponential growth policy**: $\pi_{\text{exp}} : n_t = n_0(1 + u)^t, \quad u > 0, t = 0, 1, \cdots$ .

For the exponential policy, we can identify a specific policy that achieves an exponential convergence rate as follows.

Table 1: A summarization of all experimental setups and implementations.

| Experiment | Setup Type | Generator | Reward | Dataset |
|---|---|---|---|---|
| Image Denoising | Theoretical Validation | DPM | PSNR | MNIST (Deng, 2012) |
| Math Reasoning | Practical Scenario | LLM | Accuracy | GSM-Symbolic (Mirzadeh et al., 2025) |

**Theorem 4.3** (Convergence Rate for the Exponential Policy). *Under Assumptions B.1 to B.3, there exists an exponential growth policy $\pi_{\exp}^*$ such that with probability at least $\left(1 - \sum_{t=0}^{T-1} n_t^{-4}\right)$,*

$$r^* - r\left(\theta_{\pi_{\exp}^*}^{(T)}\right) = \mathcal{O}\left((1+\zeta)^{-2T}\right),$$

*where $\zeta > 0$ is a constant defined in Theorem B.6.*

This theorem also applies to noisy rewards, as discussed in Appendix B.5.

Notice that the above three theorems analyze the convergence of the respective policies (constant, polynomial, and exponential) as the number of iterations $T$ grows without imposing any limit on the total computational budget. We now introduce the budget constraint and compare the exponential policy with the polynomial policy. Specifically, for any policy $\pi$, we introduce

$$T^*(\pi, \varepsilon) = \min_T \left\{ T \mid r^* - r\left(\theta_\pi^{(T)}\right) \leq \varepsilon \right\}, \tag{2}$$

denoting the minimum number of iterations needed for the policy $\pi$ to achieve a reward that is within $\varepsilon$ of the optimal $r^*$. The cost $C(\pi, T^*(\pi, \varepsilon))$ is then the minimum cost to attain the target performance of the policy $\pi$. The following theorem compares the total cost incurred by the exponential growth policy and polynomial policies in a worst-case scenario.

**Theorem 4.4** (Worst-Case Optimality of the Exponential Policy, informal). *For any constant or polynomial growth policy $\pi$ and a sufficiently small $\varepsilon$, we have*

$$\sup\ C(\pi_{\exp}^*, T^*(\pi_{\exp}^*, \varepsilon)) < \sup\ C(\pi, T^*(\pi, \varepsilon))$$

*with probability at least $\left(1 - \sum_{t=0}^{T^*(\pi_{\exp}^*, \varepsilon)-1} n_t^{-4}\right)$. Here, the supremum is taken over all feasible problem settings, including parameter spaces, loss functions, and reward functions under mild assumptions.*

For the formal version, please refer to Theorem B.7. This theorem shows that, in a worst-case analysis, the exponential policy $\pi_{\exp}^*$ from Theorem B.6 incurs a total cost no greater than that of any polynomial growth or constant policy to reach a specified performance threshold, establishing its efficiency and robustness. This analysis theoretically highlights the exponential policy as the preferred choice for our iterative learning tasks.

In summary, we draw the following conclusion:

- **Constant policies** lead to rewards that are consistently below optimal by a constant margin.

- **Increasing policies** such as polynomial and exponential ones can achieve optimal performance.

- In the worst case, there exists an **exponential growth policy** ensuring the same performance at equal or lower computational cost compared to any constant or polynomial growth policy.

## 5 Experiments

Our theoretical results show that, in iterative bootstrapping, synthetic data curation with an exponential scheme outperforms both linear and polynomial schemes, while linear and polynomial schemes outperform the constant scheme. In our experiments, we evaluate constant, linear (as a simplified polynomial scheme), and exponential schemes.

We conduct two experiments to evaluate our schemes: image denoising with DPMs, and mathematical reasoning with LLMs, covering both image and text domains. The setups and implementation details are summarized in Table 1.

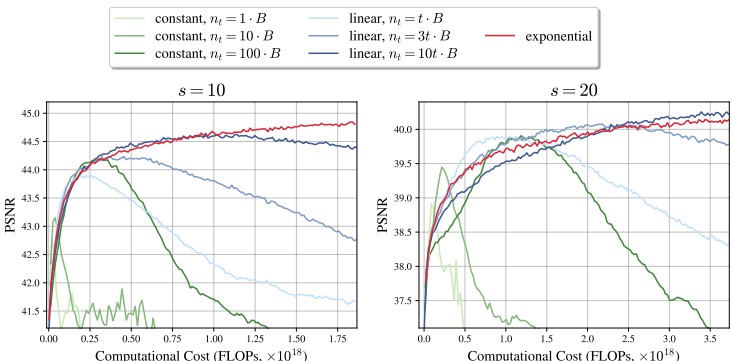

Figure 3: **Empirical results of image denoising.** The figure shows the average PSNR of the generated denoised images as a function of computational cost. Computational cost is measured in floating-point operations (FLOPs) during iterative learning, including both generation and training. The training batch size is set to $B = 640$. We adopt $n_t = 1.1^t \cdot B$ and $n_t = 1.05^t \cdot B$ as the exponential growth policies for $s = 10$ and $s = 20$, respectively.

In these settings, we emphasize that our goal is not to propose a novel task-solving method or generation technique. Rather, we focus on optimizing the allocation of generation and training costs in iterative learning with synthetic data—an objective that is method-agnostic and can be applied to any generation approach. Here, we validate our theoretical results from Section 4 and conduct a performance comparison among exponential, linear, and constant schemes on image data (using DPMs) and text data (with LLMs).

## 5.1  Image Denoising with Diffusion Models

We consider applying iterative learning to the image denoising task, a representative problem in image processing, where the goal is to recover $x$ from a noisy observation $y = x + \eta$ with an unknown Gaussian noise $\eta$.

**Setup.**  We choose diffusion probabilistic models (DPMs) (Ho et al., 2020; Song et al., 2021) as denoisers due to their intrinsic design, which aligns closely with the denoising process (Xie et al., 2023) since the reverse process of a DPM gradually estimates and removes this noise using a trained network, effectively performing denoising. Specifically, we fine-tune a pre-trained diffusion model[3] for denoising on the MNIST (Deng, 2012) dataset, and readers may refer to Appendix C for more details.

We follow the iterative learning framework to annotate synthetic "clean" images to improve the denoising process. In each iteration, the diffusion model first generates synthetic denoised images $\hat{x}$ from the noisy observations. These are then filtered with a reward model linear to the Peak Signal-to-Noise Ratio (PSNR) values compared with the real images. The filtered synthetic data are then used to fine-tune the model through a standard noise-prediction training procedure, iteratively improving its denoising capability. Although an oracle reward model may not be practical in many real-world scenarios, its use here helps to validate the generality of our theoretical results. Furthermore, existing models or correction functions with prior knowledge could be utilized as the reward, as demonstrated in Gillman et al. (2024).

**Implementations.**  We perform denoising for two noise levels corresponding to the diffusion step $s = 10$ and 20, where the PSNR of noisy images are 33.25 and 28.40, respectively. A summary of the algorithm is provided in Algorithm 2. We consider three constant and three linear policies, each with small, medium, and large $n_0$. For the exponential policy, we have performed a small hyperparameter search over $n_0$ and $u$, selecting good configurations for two settings. The complete results of the hyperparameter search are presented as an ablation study on different exponential schemes in Appendix F.2, demonstrating its robustness. The configurations for all schemes are presented in the caption and legend of Figure 3.

---

[3] `https://huggingface.co/google/ddpm-cifar10-32`

Table 2: **Comparison of each policy on the image denoising and math reasoning tasks.** The final accuracy are determined using a held-out validation set. We use **boldface** for the best result while the underline for the second best. For the exponential policies, we only exhibit a single set of configurations for the comparison. We plot for all the policies in Figure 3 and Figure 4, respectively.

<table>
<tr><td colspan="4" align="center">(a) Image denoising</td><td colspan="4" align="center">(b) Math reasoning</td></tr>
<tr><td>POLICY</td><td>$n_t$</td><td>$s = 10$</td><td>$s = 20$</td><td>$n_t$</td><td>SYMBOLIC</td><td>P1</td><td>P2</td></tr>
<tr><td>PRE-TRAINED</td><td>N/A</td><td>40.92</td><td>36.96</td><td>N/A</td><td>55.60</td><td>34.98</td><td>15.72</td></tr>
<tr><td rowspan="3">CONSTANT</td><td>$1 \cdot B$</td><td>42.75</td><td>38.91</td><td>$10 \cdot B$</td><td>60.28</td><td>38.50</td><td>16.93</td></tr>
<tr><td>$10 \cdot B$</td><td>43.15</td><td>39.45</td><td>$30 \cdot B$</td><td>60.16</td><td>39.68</td><td>17.73</td></tr>
<tr><td>$100 \cdot B$</td><td>44.18</td><td>39.91</td><td>$100 \cdot B$</td><td>64.04</td><td>41.66</td><td>18.81</td></tr>
<tr><td rowspan="3">LINEAR</td><td>$t \cdot B$</td><td>43.90</td><td>39.90</td><td>$3t \cdot B$</td><td>62.54</td><td>38.88</td><td>17.01</td></tr>
<tr><td>$3t \cdot B$</td><td>44.23</td><td>40.08</td><td>$10t \cdot B$</td><td>61.96</td><td>39.94</td><td>17.29</td></tr>
<tr><td>$10t \cdot B$</td><td>44.64</td><td>**40.26**</td><td>$30t \cdot B$</td><td>64.24</td><td>40.84</td><td>19.17</td></tr>
<tr><td>EXPONENTIAL</td><td>—</td><td>**44.84**</td><td>40.14</td><td>—</td><td>**65.66**</td><td>**47.26**</td><td>**20.65**</td></tr>
</table>

**Results.** We summarize the final performance of different schemes and baselines in Table 2 and show the trade-off between performance and computational cost in Figure 3. We observe that the exponential policy yields the best performance for $s = 10$, and performs comparable to the best linear policy for $s = 20$. This observation demonstrates both the benefits of an increasing policy (Theorem 4.2) and the robustness of the exponential growth policy (Theorem 4.4). In contrast, the constant policy only matches the exponential policy in the early stages of training, but soon reaches a performance plateau and eventually decays due to overfitting, corroborating Theorem 4.1.

## 5.2 Math Reasoning with LLMs

Now we move to natural language processing. Iterative learning with synthetic data is also an emerging method for LLMs. Previous research (Zelikman et al., 2022) leverages the model to generate synthetic data, which is then filtered using a reward model or accuracy metrics and subsequently utilized to fine-tune the model. This framework is employed in various settings, including instruction tuning (Touvron et al., 2023), alignment (Dong et al., 2023), and recent efforts to enhance reasoning capabilities through chain-of-thought (CoT) approaches (Hosseini et al., 2024; Lehnert et al., 2024; Singh et al., 2025).

**Setup.** We select mathematical reasoning to evaluate the effectiveness of our strategy in iterative learning. Let $q$ and $a$ represent the question and its corresponding answer, respectively. We use the LLM to generate synthetic solutions with CoT and in-context examples. These synthetic data are filtered according to the correctness of the final generated answer $\hat{a}$, defined as the reward $\mathcal{R}(\hat{a}) = \mathbf{1}(\hat{a} = a)$. The filtered data are used to further train the model in an iterative process, without filtering based on the correctness of CoT itself. In summary, synthetic data with the correct final solutions are generated and used for training in each iteration, closely mimicking the STaR method (Zelikman et al., 2022).

**Implementations.** Many contemporary LLMs are known to overfit to GSM8k (Cobbe et al., 2021) or exhibit a certain degree of data leakage (Zhang et al., 2024; Mirzadeh et al., 2025), where even minor changes to numbers or names can result in significant performance degradation. This issue renders the evaluation results on GSM8k unreliable. To address this, we regenerate three math datasets—GSM_symbolic, GSM_p1, and GSM_p2—using symbolic math templates from Mirzadeh et al. (2025), one of the most outstanding papers on evaluating math reasoning. Each dataset consists of questions paired with their correct answers. GSM_symbolic retains a similar difficulty level to GSM8k, while GSM_p1 and GSM_p2 introduce one and two additional clauses per question, respectively, making them progressively more challenging. Together, these datasets complete a robust evaluation set.

In our iterative learning framework, we generate synthetic data using a mixing ratio of 7:2:1 across these three datasets, filter the data based on the correctness of the final answers, and fine-tune the model on the selected data. We use the Llama-3-8B-Base model (Dubey et al., 2024) with full-

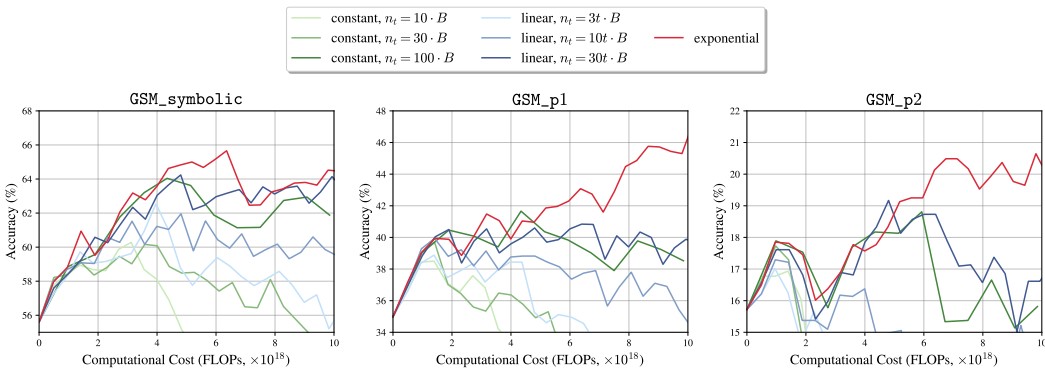

Figure 4: **Empirical results of math reasoning.** The figure shows the accuracies with respect to the computational cost, measured in FLOPs as well. The training batch size is set to $B = 256$. We adopt $n_t = 10 \cdot 2^t \cdot B$ as the exponential policy. We set the temperature to 0.3 during the generation step and include the results for a temperature of 0.7 in Appendix F.4, which lead to the same conclusions.

weight fine-tuning. For evaluation, the model is tested on held-out datasets from all three difficulty levels. During data generation and evaluation, in-context examples and CoT reasoning are used, but in-context examples are excluded during fine-tuning.

A summary of the algorithm is provided in Algorithm 3. Similarly to image denoising, we consider three constant and three linear policies, each with small, medium, and large $n_0$. For the exponential scheme, we have selected one configuration and left the full ablation results in Appendix F.3. The configurations for all schemes are detailed in Figure 4 and its caption. We also include the performance of the pre-trained model as a baseline.

**Results.** Figure 4 shows the performance versus computational cost across the three datasets under iterative training, while Table 2 summarizes the final performance for all policies. Across all three difficulty levels, the exponential growth policy demonstrates the steadiest improvement compared to other schemes. Although the linear policy ranks second overall, it fails to match the exponential growth policy when the training budget is large. In particular, with the exponential growth policy, iterative learning increases accuracy on GSM_p1 from 35% to 47% and on GSM_p2 from 16% to 21%. Based on these results, we recommend the exponential scheme for iterative learning.

## 6 Conclusion

In this work, we take the first step toward understanding how to design optimal budget allocation policies in iterative bootstrapping for supervised fine-tuning. Through a combination of theoretical analysis and empirical validations, we demonstrate that constant policies are insufficient for achieving convergence, whereas increasing policies offer a more effective alternative. Among increasing policies, the exponential growth policy emerges as a robust and efficient method.

Although our focus here is on SFT in an iterative learning framework, recent research has highlighted the potential of Reinforcement Learning with Human Feedback (RLHF) (Yuan et al., 2024; Pang et al., 2024; Setlur et al., 2024) for similar iterative setups. Extending our framework to incorporate RLHF is a promising direction for future work. By answering these open questions, we aim to lay a foundation for more efficient iterative training approaches that can improve the post-training of foundation models.

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

# A Theoretic Intuition

## A.1 Discussion on the Gaussian Setting

In the Gaussian case, we use a simplified toy example that omits the generation cost only to provide a clear, introductory setup for the reader. In our complete framework (see Section 4), however, we assume a generation cost for standard estimators and incorporate this cost into the objective using the hyperparameter $c_g$. This hyperparameter offers flexibility: setting $c_g = 0$ corresponds to free generation, while a large $c_g$ reflects expensive generation. We also provide general theorems that hold for all values of $c_g$, and explain how the Gaussian case satisfies all assumptions of the general theorems in Appendix B.2. Thus, the Gaussian case can be viewed as a special instance, and the theoretical results still remain valid regardless of whether the sampling is free or costly.

## A.2 Proof of Theorem 3.1

We first calculate the distribution of $\theta^{(T)}$ given an initial parameter $\theta^{(0)}$ and a specific policy $\pi : \{n_t\}_t$ with the terminal time $T$ in the following lemma.

**Lemma A.1.** *With the MLE updation and Gaussian distributions, given an initial parameter $\theta^{(0)}$ and a specific policy $\pi : \{n_t\}_t$ with the terminal time $T$, we have*

$$p(\theta^{(T)} \mid \theta^{(0)}, \pi) = \mathcal{N} \left( \frac{\theta^{(0)}}{(1 + \sigma^2/\kappa^2)^T}, \sigma^2 \sum_{t=0}^{T-1} \frac{1}{n_t(1 + \sigma^2/\kappa^2)^{2(T-t)-1}} \right).$$

*Proof.* Suppose that there are $N_t$ data $\{x_i\}_{i=1}^{N_t}$ generated from the $t$-th step generator such that the selected sample size is $n_t$. Let $b_i \sim \text{Bernoulli}(\mathcal{R}(x_i))$ as a selecting sign where $b_i = 1$ indicates that $x_i$ is selected and placed into $D_t$, otherwise $b_i = 0$ indicates that $x_i \notin D_t$.

We first consider one-step updating. We have the distribution of a data $x_i$ conditioned on $x_i \in D_t$,

$$
\begin{aligned}
p\left(x_i \mid x_i \in D_t\right) &= p\left(x_i \mid b_i = 1\right) \\
&\propto \mathcal{R}(x_i) \cdot \mathcal{N}\left(x_i; \theta^{(t)}, \sigma^2\right) \\
&\propto \exp\left(-\frac{x_i^2}{2\kappa^2} - \frac{(x_i - \theta^{(t)})^2}{2\sigma^2}\right) \\
&\sim \mathcal{N}\left(\frac{\theta^{(t)}}{1 + \sigma^2/\kappa^2}, \frac{\sigma^2}{1 + \sigma^2/\kappa^2}\right).
\end{aligned}
$$

Since $\theta^{(t+1)}$ is the solution of the MLE satisfying

$$\theta^{(t+1)} = \arg\max_\theta \prod_{x \in D_t} \mathcal{N}(x; \theta, \sigma^2) = \frac{1}{n} \sum_{x \in D_t} x_i,$$

we have the distribution of $\theta^{(t+1)}$ conditioned on the parameter $\theta^{(t)}$ and the sample size $n_t$ of the previous iteration, written as

$$\theta^{(t+1)} \mid \theta^{(t)}, n_t \sim \mathcal{N}\left(\frac{\theta^{(t)}}{1 + \sigma^2/\kappa^2}, \frac{\sigma^2}{n_t(1 + \sigma^2/\kappa^2)}\right).$$

Considering the decomposition

$$\theta^{(t+1)} = \frac{\theta^{(t)}}{1 + \sigma^2/\kappa^2} + \frac{\sigma}{n_t^{1/2}(1 + \sigma^2/\kappa^2)^{1/2}} \xi_t,$$

with an independent auxiliary random $\xi_t \sim \mathcal{N}(0, 1)$ recursively, we have

$$\theta^{(T)} = \frac{\theta^{(0)}}{(1 + \sigma^2/\kappa^2)^T} + \sum_{t=0}^{T-1} \frac{\sigma}{n_t^{1/2}(1 + \sigma^2/\kappa^2)^{1/2 + (T-1-k)}} \xi_t.$$

Therefore,

$$\theta^{(T)} \mid \theta^{(0)}, \pi \sim \mathcal{N}\left(\frac{\theta^{(0)}}{(1 + \sigma^2/\kappa^2)^T}, \sum_{t=0}^{T-1} \frac{\sigma^2}{n_t(1 + \sigma^2/\kappa^2)^{2(T-t)-1}}\right).$$

$\square$

Back to the proof of the theorem, we define the mean and variance as

$$\mu_T = \frac{\theta^{(0)}}{(1 + \sigma^2/\kappa^2)^T},$$

and

$$\sigma_T^2 = \sigma^2 \sum_{t=0}^{T-1} \frac{1}{n_t(1 + \sigma^2/\kappa^2)^{2(T-t)-1}}.$$

According to Lemma A.1, we have that

$$\begin{aligned}
\mathbb{E}_{\theta^{(T)}} \mathbb{E}_x[\mathcal{R}(x)] &= \mathbb{E}_{\theta^{(T)}} \mathbb{E}_x\left[\exp\left(-\frac{x^2}{2\kappa^2}\right)\right] \\
&= \mathbb{E}_{\theta^{(T)}}\left[\int \frac{1}{\sqrt{2\pi\sigma^2}} \exp\left(-\frac{x^2}{2\kappa^2} - \frac{(x - \theta^{(T)})^2}{2\sigma^2}\right) dx\right] \\
&= \frac{1}{\sqrt{1 + \sigma^2/\kappa^2}} \mathbb{E}_{\theta^{(T)}}\left[\exp\left(-\frac{(\theta^{(T)})^2}{2(\sigma^2 + \kappa^2)}\right)\right] \\
&= \frac{1}{\sqrt{1 + \sigma^2/\kappa^2}} \frac{1}{\sqrt{1 + \sigma_T^2/(\sigma^2 + \kappa^2)}} \exp\left(-\frac{\mu_T^2}{2(\sigma^2 + \kappa^2 + \sigma_T^2)}\right) \\
&= \frac{\kappa}{\sqrt{\sigma^2 + \kappa^2 + \sigma_T^2}} \exp\left(-\frac{\mu_T^2}{2(\sigma^2 + \kappa^2 + \sigma_T^2)}\right).
\end{aligned}$$

With the assumption that $\mu_T \leq \sqrt{\sigma^2 + \kappa^2}$, the right term of the above equation is an increasing function with respect to $\frac{1}{\sqrt{\sigma^2 + \kappa^2 + \sigma_T^2}}$. Therefore, the objective function gets its maximum value when we minimize $\sigma_T^2$. Furthermore, we have the optimal iterative learning policy from Cauchy–Schwarz inequality,

$$\begin{aligned}
\sigma_T^2 \cdot \sum_{t=0}^{T-1} n_t &= \sigma^2 \left(\sum_{t=0}^{T-1} \frac{1}{n_t(1 + \sigma^2/\kappa^2)^{2(T-t)-1}}\right) \cdot \left(\sum_{t=0}^{T-1} n_t\right) \\
&\geq \sigma^2(1 + \sigma^2/\kappa^2)\left(\sum_{t=0}^{T-1} \frac{1}{(1 + \sigma^2/\kappa^2)^{T-t}}\right)^2,
\end{aligned}$$

where the equal sign is established if and only if

$$n_t \propto \left(1 + \frac{\sigma^2}{\kappa^2}\right)^{t-T}.$$

Reindexing or shifting $t$ will complete the proof.

# B   Main Theoretic Results

## B.1   Notations

Consider a probability space $\mathcal{S} = (\Omega, \mathcal{B}(\Omega), \mathcal{P})$, where $\Omega$ is the sample space, $\mathcal{B}(\Omega)$ is the Borel $\sigma$-algebra on $\Omega$, and $\mathcal{P}$ is the probability measure. Let $\Theta \subseteq \mathbb{R}^p$ be a parameter space, and define

$$\mathcal{A}_\theta = \{f(\omega; \theta) \mid \theta \in \Theta\}$$

as a parametric family of random variables mapping from $\Omega$ to a measurable outcome space $\mathcal{T}$. Each $\theta$ induces a distribution $\mathcal{P}_\theta$ on $\mathcal{T}$, where

$$\mathcal{P}_\theta(X) \;=\; \mathcal{P}\left(f(\omega;\theta) \in X\right) \quad \text{for any measurable set } X \subseteq \mathcal{T}.$$

Any function $\mathcal{R} : \mathcal{T} \to [0,1]$ induces an additional distribution $\mathcal{P}_{r;\theta}$ on $[0,1]$, where

$$\mathcal{P}_{\mathcal{R};\theta}(\mathcal{B}) \;=\; \mathcal{P}_\theta\left(\mathcal{R}(x) \in \mathcal{B}\right) \quad \text{for any Borel set } \mathcal{B} \subseteq [0,1].$$

Given a reward model $\mathcal{R}(x)$, the reward $r(\theta)$ for $\theta \in \Theta$ is defined as the expectation of the distribution $\mathcal{P}_{\mathcal{R};\theta}$, i.e.,

$$r(\theta) := \int_{[0,1]} r \; \mathrm{d}\mathcal{P}_{\mathcal{R};\theta}(r) = \mathbb{E}_{x \sim \mathcal{P}(\theta)}[\mathcal{R}(x)].$$

Define the loss function $l(x;\theta) : \mathcal{T} \times \Theta \to [0,\infty)$. Now we consider the optimizing algorithm that starts with an initial parameter $\theta^{(0)}$ and updates the parameter $\theta^{(t)}$ at the $t$-th iteration using gradient descent with a learning rate $\eta > 0$ and data $D_t \in \mathcal{T}^{\otimes |D_t|}$, formulated as

$$\theta^{(t+1)} = \theta^{(t)} - \frac{\eta}{n_t} \sum_{x \in D_t} \nabla_\theta l\left(x;\theta^{(t)}\right), \quad t = 0, 1, \cdots.$$

## B.2   Assumptions

We always consider the general case where the initial value $\theta^{(0)}$ is sufficiently accurate. The distribution $\mathcal{P}_{\theta^{(0)}}$ is assumed as non-degenerate and there exists a constant $c_r$ such that

$$\mathrm{d}\mathcal{P}_{r;\theta^{(0)}} \geq c_r > 0.$$

**Assumption B.1** (*Basic properties of the optimization space and loss function*). The parameter space $\Theta$ for $\theta$ and the loss function $l(x;\theta)$ indexed by $\theta \in \Theta$ satisfy the following requirements.

1. The covering number $\mathcal{N}(\epsilon, \Theta, \|\cdot\|_2)$ for the parameter space $\Theta$ is bounded, which means the inequality

$$\mathcal{N}(\epsilon, \Theta, \|\cdot\|_2) \leq c_\Theta \varepsilon^{-C_\Theta} \vee 1, \forall \epsilon > 0 \tag{3}$$

   holds for some positive constants $c_\Theta$ and $C_\Theta$.

2. The loss function $l(x;\theta)$ is Lipschitz continuous. Formally, there exists a constant $L > 0$ independent of $x$ such that

$$|l(x;\theta) - l(x;\theta_1)| \leq C_L \|\theta - \theta_1\|_2, \forall \theta, \theta_1 \in \Theta. \tag{4}$$

3. For any $\theta \in \Theta$, there exist positive constants $c$ and $\beta$ satisfying

$$\|\nabla_\theta l(x;\theta)\|_2^2 \geq cl(x;\theta), \quad \left\|\nabla_{\theta\theta'}^2 l(x;\theta)\right\|_{\mathrm{op}} \leq \beta. \tag{5}$$

*Remark.* This assumption governs both the complexity of the parameter space $\Theta$ and the smoothness of the likelihood function $l(x;\theta)$. Here, we provide two examples to show that these assumptions are reasonable. Firstly, if $\Theta$ is a compact finite-dimensional space or a functional space sufficiently smooth, then the coverage number requirement (Equation (3)) is satisfied. Secondly, for a special case when $l(x;\theta)$ is the negative log-likelihood for a Gaussian distribution, the derivative condition (Equation (5)) holds with $c = 2$ and $\beta = 1$.

We define the *post-selection distribution* $\mathcal{Q}_\theta$ via its Radon–Nikodým derivative relative to $\mathcal{P}_\theta$ as

$$\frac{\mathrm{d}\mathcal{Q}_\theta(x)}{\mathrm{d}\mathcal{P}_\theta(x)} \;=\; \frac{\mathcal{R}(x)}{\mathbb{E}_{x \sim \mathcal{P}_\theta}[\mathcal{R}(x)]},$$

which indicates that $\mathcal{Q}_\theta$ is just the original distribution $\mathcal{P}_\theta$ weighted by the reward function $\mathcal{R}(x)$. Equivalently, we can think of $\mathcal{Q}_\theta$ as the conditional distribution of $x$ given that $x$ is "selected" in proportion to its reward. For convenience, let $p_\theta$, $q_\theta$, and $p_{r;\theta}$ be the density functions (with respect to some appropriate base measure) of $\mathcal{P}_\theta$, $\mathcal{Q}_\theta$, and $\mathcal{P}_{r;\theta}$, respectively.

Next, we define the *expected likelihood function* $\mathbb{P}l_{\theta_1}(\theta_2)$ as

$$\mathbb{P}l_{\theta_1}(\theta_2) = \mathbb{E}_{x \sim \mathcal{Q}_{\theta_1}}\left[l(x;\theta_2)\right] = \mathbb{E}_{x \sim q_{\theta_1}}\left[-\log p_{\theta_2}(x)\right].$$

Intuitively, $\mathbb{P}l_{\theta_1}(\theta_2)$ measures how well the model $p_{\theta_2}$ "fits" data drawn from the post-selection distribution $\mathcal{Q}_{\theta_1}$.

We define

$$\Theta^* = \arg\min_{\theta \in \Theta} \mathbb{P}l_\theta(\theta)$$

as the *optimal parameter set* that minimize all possible $\mathbb{P}l_\theta(\theta)$.

Finally, to introduce the sampling error, we define

$$\mathbb{P}_n l(X;\theta) = \frac{1}{n}\sum_{i=1}^{n} l\left(x_i;\theta\right)$$

as an empirical evaluation for the negative log-likelihood based on samples $X = (x_1, \ldots, x_n)$.

**Assumption B.2** (*Local reducibility of the expected loss function*). There exists a constant $\gamma \in (0,1]$ such that for any $\theta, \theta_1 \in \Theta$ with $\|\theta - \theta_1\|_2 \leq 2C_L/\beta$ and $\mathbb{P}l_{\theta_1}(\theta) \leq \mathbb{P}l_{\theta_1}(\theta_1)$, we have

$$\mathbb{P}l_\theta(\theta) - \mathbb{P}l_{\theta_1}(\theta_1) \leq \gamma\left(\mathbb{P}l_{\theta_1}(\theta) - \mathbb{P}l_{\theta_1}(\theta_1)\right).$$

*Remark.* This assumption ensures that, within a local neighborhood of $\theta$, any parameter update that reduces the expected likelihood under $\theta_1$'s distribution also decreases the expected likelihood under $\theta$'s distribution, relative to $\mathbb{P}l_{\theta_1}(\theta_1)$. Concretely, each iteration of the algorithm samples from $\theta_1$ but then updates the parameter $\theta$. Since the data remain the same while $\theta$ changes, this condition guarantees that a reduction in the loss is preserved across iterations. For instance, if $\mathcal{P}_\theta$ is a Gaussian distribution with mean $\theta$ and standard deviation 1, and $\mathcal{R}(x)$ is the normalized standard Gaussian density, one can verify that Assumption B.2 holds with $\gamma = 1/2$.

**Assumption B.3** (*Relationship between the loss and the reward function*). For any $\theta \in \Theta$, it holds for some $c_2, c_3 \leq 1$ that

$$c_3\mathbb{P}l_\theta(\theta) \leq r^* - r(\theta) \leq c_2\mathbb{P}l_\theta(\theta).$$

*Remark.* This assumption establishes a direct relationship between the performance gap $(r^* - r(\theta))$ and the expected loss $\mathbb{P}l_\theta(\theta)$. Specifically, the upper bound shows that if $\mathbb{P}l_\theta(\theta)$ is small, then $(r^* - r(\theta))$ must also be small, which is crucial for proving convergence of our proposed policy. For example, in the Gaussian setting discussed in the remark of Assumption B.2, one can verify that $c_2 = 1/12$. Conversely, the lower bound indicates that a small performance gap necessarily implies a correspondingly small expected loss, thereby capturing the sensitivity of the loss to improvements in the expected reward. Since both $(r^* - r(\theta))$ and $\mathbb{P}l_\theta(\theta)$ quantify how well the policy performs, it is natural that each can bound the other. This relationship serves as a key underpinning for the theoretical results presented in subsequent sections.

### B.3 Main Theorems

Our theoretical results start from the defect of constant policies by showing that the final reward remains bounded away from the optimum with non-negligible probability when using constant policies.

**Theorem B.4** (Bounded Reward for Constant policy). *Under Assumption B.3, there exists a constant $c > 0$ such that, for any $T$ and any constant policy $\pi_{\text{const}}$, with probability at least $1/4$,*

$$r^* - r\left(\theta_{\pi_{\text{const}}}^{(T)}\right) \geq cn_0^{-1/2}.$$

On the contrary, any increasing policy eventually closes the optimization gap within arbitrarily small range as long as the training iteration goes to infinity.

**Theorem B.5** (Optimal Reward for Increasing Policies). *Under Assumptions B.1 to B.3, for any increasing policy $\pi$ and $\varepsilon > 0$, there exists a sufficiently large $T$ such that, with probability greater than $\left(1 - \sum_{t=0}^{T-1} n_t^{-4}\right)$,*

$$r^* - r\left(\theta_\pi^{(T)}\right) \leq \varepsilon.$$

Next, we demonstrate that the exponential-growth policy from Theorem B.6 achieves this convergence more effectively than polynomial growth policies, including the constant policies.

**Theorem B.6** (Convergence Rate for Exponential Policy). *Under Assumptions B.1 to B.3, there exists an exponential growth policy $\pi_{\exp}^*$ such that with probability at least $\left(1 - \sum_{t=0}^{T-1} n_t^{-4}\right)$,*

$$r^* - r\left(\theta_{\pi_{\exp}^*}^{(T)}\right) = \mathcal{O}\left((1+\zeta)^{-2T}\right)$$

*with a constant $\zeta = \gamma c/(8\beta - \gamma c) > 0$.*

Consider the minimum number of iterations $T^*(\pi, \varepsilon)$ needed for a policy $\pi$ to achieve a reward that is within $\varepsilon$ of the optimal $r^*$ as defined in Equation (2), then the cost $C(\pi, T^*(\pi, \varepsilon))$ is the minimum cost to attain the target performance for policy $\pi$. The following theorem compares the total cost incurred by exponential growth policies and polynomial growth policies (including constant policies) in a worst-case scenario.

**Theorem B.7** (Worst-Case Optimality of Exponential Policy). *Under the same conditions as in Theorem B.6, for any constant or polynomial growth policy $\pi$ and a sufficiently small $\varepsilon$, we have*

$$\sup_{\Theta, \Theta^*, l(x;\theta), \mathcal{R}(x)} C\left(\pi_{\exp}^*, T^*(\pi_{\exp}^*, \varepsilon)\right) \leq \sup_{\Theta, \Theta^*, l(x;\theta), \mathcal{R}(x)} C\left(\pi, T^*(\pi, \varepsilon)\right)$$

*with probability at least $\left(1 - \sum_{t=0}^{T^*(\pi_{\exp}, \varepsilon)-1} n_t^{-4}\right)$. The supremum is taken over all possible parameter spaces $\Theta, \Theta^*$, loss functions $l(x;\theta)$, and reward models $\mathcal{R}(x)$ satisfying the same conditions as in Theorem B.6.*

Such a worst-case perspective shows that the exponential growth policy remains robust and outperforms (or at least matches) any constant or polynomial policy. (Although the policy $\pi$ is fixed in advance, the iterative sequence $\left\{\theta_\pi^{(t)}\right\}_{t=0}^{T-1}$ is still random because data are sampled and selected probabilistically at each iteration. The stated probability bound reflects this inherent randomness and guarantees high-probability performance under the specified assumptions.

## B.4 Proofs

In this subsection, we present proofs for all statements in our theoretical results. We begin with the proof of Theorem B.6, proceed to prove Theorems B.4 and B.5, and conclude with the proof of Theorem B.7.

*Proof of Theorem B.6.*

We begin by noting the following decomposition

$$\begin{aligned}
\mathbb{P}l_{\theta^{(0)}}\left(\theta^{(1)}\right) - \mathbb{P}l_{\theta^{(0)}}\left(\theta^{(0)}\right) = {}& \mathbb{P}l_{\theta^{(0)}}\left(\theta^{(1)}\right) - \mathbb{P}_{n_0}l\left(X;\theta^{(1)}\right) \\
& + \mathbb{P}_{n_0}l\left(X;\theta^{(1)}\right) - \mathbb{P}_{n_0}l\left(X;\theta^{(0)}\right) \\
& + \mathbb{P}_{n_0}l\left(X;\theta^{(0)}\right) - \mathbb{P}l_{\theta^{(0)}}\left(\theta^{(0)}\right),
\end{aligned}$$

which gives

$$\begin{aligned}
\mathbb{P}l_{\theta^{(0)}}\left(\theta^{(1)}\right) - \mathbb{P}l_{\theta^{(0)}}\left(\theta^{(0)}\right) \leq {}& \mathbb{P}_{n_0}l\left(X;\theta^{(1)}\right) - \mathbb{P}_{n_0}l\left(X;\theta^{(0)}\right) \\
& + \left|\mathbb{P}l_{\theta^{(0)}}\left(\theta^{(1)}\right) - \mathbb{P}_{n_0}l\left(X;\theta^{(1)}\right)\right| + \left|\mathbb{P}l_{\theta^{(0)}}\left(\theta^{(0)}\right) - \mathbb{P}_{n_0}l\left(X;\theta^{(0)}\right)\right|.
\end{aligned} \tag{6}$$

By Assumption B.1 and a Taylor expansion, choosing the step size $\eta = 1/(2\beta)$ gives

$$
\begin{aligned}
\mathbb{P}_{n_0} l\left(X; \theta^{(1)}\right) - \mathbb{P}_{n_0} l\left(X; \theta^{(0)}\right) &\leq \left(\nabla_\theta \mathbb{P}_{n_0} l\left(X; \theta^{(0)}\right)\right)^\mathsf{T} \left(\theta^{(1)} - \theta^{(0)}\right) + \beta \left\|\theta^{(1)} - \theta^{(0)}\right\|_2^2 \\
&\leq -\eta \left(\nabla_\theta \mathbb{P}_{n_0} l\left(X; \theta^{(0)}\right)\right)^\mathsf{T} \nabla_\theta \mathbb{P}_{n_0} l\left(X; \theta^{(0)}\right) \\
&\qquad\qquad + \beta\eta^2 \left\|\nabla_\theta \mathbb{P}_{n_0} l\left(X; \theta^{(0)}\right)\right\|_2^2 \\
&\leq -\frac{1}{4\beta} \left\|\nabla_\theta \mathbb{P}_{n_0} l\left(X; \theta^{(0)}\right)\right\|_2^2 \\
&\leq -\frac{c}{4\beta} \mathbb{P}_{n_0} l\left(X; \theta^{(0)}\right).
\end{aligned}
\tag{7}
$$

Define

$$
\mathcal{R}_\theta = \mathbb{E}_\varepsilon \left[\sup_{\theta \in \Theta} \left|\frac{1}{n} \sum_{i=1}^n \varepsilon_i l(x_i; \theta)\right|\right],
$$

where $\varepsilon_i$ $(1 \leq i \leq n)$ are i.i.d. Rademacher random variables. By Dudley's entropy integral and Assumption B.1, we then get

$$
\begin{aligned}
\sup_{\theta \in \Theta} \left|\mathbb{P} l_{\theta^{(0)}}(\theta) - \mathbb{P}_n l(X; \theta)\right| &\leq 2\mathcal{R}_\theta \\
&\leq 2L n^{-1/2} \int_0^\infty \sqrt{\log \mathcal{N}(\epsilon, \Theta, \|\cdot\|_2)}\, d\epsilon \tag{8} \\
&\leq 2LC_\Theta \log c_\Theta n^{-1/2}.
\end{aligned}
$$

By Chernoff-Hoeffding theorem,

$$
P\left(\left|n_0 - N_0 r\left(\theta^{(0)}\right)\right| \geq t_1\right) \leq \exp\left(\frac{-2t_1^2}{N_0 r\left(\theta^{(0)}\right)}\right).
$$

Combining Equations (7) and (8) with Equation (6) and choosing

$$
t_1 = 2\left(N_0 r\left(\theta^{(0)}\right)\right)^{1/2} \log N_0,
$$

we see that, with probability at least $\left(1 - N_0^{-4}\right)$,

$$
\begin{aligned}
&\mathbb{P} l_{\theta^{(0)}}\left(\theta^{(1)}\right) - \mathbb{P} l_{\theta^{(0)}}\left(\theta^{(0)}\right) \\
&\leq -\frac{c}{4\beta} \mathbb{P} l_{\theta^{(0)}}\left(\theta^{(0)}\right) + \left(2 - \frac{c}{4\beta}\right) \sup_{\theta \in \Theta} \left|\mathbb{P} l_{\theta^{(0)}}(\theta) - \mathbb{P}_{n_0} l(X; \theta)\right| \\
&\leq -\frac{c}{4\beta} \mathbb{P} l_{\theta^{(0)}}\left(\theta^{(0)}\right) + \left(2 - \frac{c}{4\beta}\right) 2LC_\Theta \log c_\Theta n_0^{-1/2} \\
&\leq -\frac{c}{4\beta} \mathbb{P} l_{\theta^{(0)}}\left(\theta^{(0)}\right) + \left(2 - \frac{c}{4\beta}\right) 2LC_\Theta \log c_\Theta \left(N_0 r\left(\theta^{(0)}\right) - 2\left(N_0 r\left(\theta^{(0)}\right)\right)^{1/2} \log N_0\right)^{-1/2}.
\end{aligned}
$$

We choose $N_0$ to satisfy

$$
\left(2 - \frac{c}{4\beta}\right) 2LC_\Theta \log c_\Theta \left(N_0 r\left(\theta^{(0)}\right) - 2\left(N_0 r\left(\theta^{(0)}\right)\right)^{1/2} \log N_0\right)^{-1/2} \leq \frac{c}{8\beta} \mathbb{P} l_{\theta^{(0)}}\left(\theta^{(0)}\right).
$$

By Assumption B.2, we have

$$
\mathbb{P} l_{\theta^{(1)}}\left(\theta^{(1)}\right) - \mathbb{P} l_{\theta^{(0)}}\left(\theta^{(0)}\right) \leq \gamma \left(\mathbb{P} l_{\theta^{(0)}}\left(\theta^{(1)}\right) - \mathbb{P} l_{\theta^{(0)}}\left(\theta^{(0)}\right)\right) \leq -\frac{\gamma c}{8\beta} \mathbb{P} l_{\theta^{(0)}}\left(\theta^{(0)}\right),
$$

which implies

$$
\mathbb{P} l_{\theta^{(1)}}\left(\theta^{(1)}\right) \leq \mathbb{P} l_{\theta^{(0)}}\left(\theta^{(0)}\right) \left(1 - \frac{\gamma c}{8\beta}\right)^1.
$$

Define $\tilde{\mathbb{P}}l_\theta = \mathbb{P}l_\theta(\theta)$. This can be rewritten as

$$\tilde{\mathbb{P}}l_{\theta^{(1)}} - \tilde{\mathbb{P}}l_{\theta^{(0)}} \leq -\frac{\gamma c}{8\beta}\tilde{\mathbb{P}}l_{\theta^{(0)}},$$

and by Assumption B.3 we have the inequality

$$r\left(\theta^{(1)}\right) \geq r^* - c_2\tilde{\mathbb{P}}l_{\theta^{(1)}} \geq r^* - c_2\tilde{\mathbb{P}}l_{\theta^{(0)}}\left(1 - \frac{\gamma c}{8\beta}\right)^1.$$

By induction, we see that

$$\mathbb{P}l_{\theta^{(t)}}\left(\theta^{(t)}\right) \leq \mathbb{P}l_{\theta^{(0)}}\left(\theta^{(0)}\right)\left(1 - \frac{\gamma c}{8\beta}\right)^t.$$

For each $t$, with probability at least $1 - N_t^{-4}$,

$$\mathbb{P}l_{\theta^{(t)}}\left(\theta^{(t+1)}\right) - \mathbb{P}l_{\theta^{(t)}}\left(\theta^{(t)}\right)$$

$$\leq -\frac{c}{4\beta}\mathbb{P}l_{\theta^{(t)}}\left(\theta^{(t)}\right) + \left(2 - \frac{c}{4\beta}\right)\sup_{\theta\in\Theta}|\mathbb{P}l_{\theta^{(t)}}(\theta) - \mathbb{P}_{n_t}l(X;\theta)|$$

$$\leq -\frac{c}{4\beta}\mathbb{P}l_{\theta^{(t)}}\left(\theta^{(t)}\right) + \left(2 - \frac{c}{4\beta}\right)2C_LC_\Theta\log c_\Theta n_t^{-1/2}$$

$$\leq -\frac{c}{4\beta}\mathbb{P}l_{\theta^{(t)}}\left(\theta^{(t)}\right) + \left(2 - \frac{c}{4\beta}\right)2C_LC_\Theta\log c_\Theta\left(N_tr\left(\theta^{(t)}\right) - 2\left(N_tr\left(\theta^{(t)}\right)\right)^{1/2}\log N_t\right)^{-1/2}.$$

We now choose $N_t$ so that

$$\left(2 - \frac{c}{4\beta}\right)2C_LC_\Theta\log c_\Theta\left(N_tr\left(\theta^{(t)}\right) - 2\left(N_tr\left(\theta^{(t)}\right)\right)^{1/2}\log N_t\right)^{-1/2} \leq \frac{c}{8\beta}\mathbb{P}l_{\theta^{(0)}}\left(\theta^{(0)}\right)\left(1 - \frac{\gamma c}{8\beta}\right)^t.$$

Then

$$\mathbb{P}l_{\theta^{(t)}}\left(\theta^{(t+1)}\right) \leq \left(1 - \frac{c}{4\beta}\right)\mathbb{P}l_{\theta^{(t)}}\left(\theta^{(t)}\right) + \frac{c}{8\beta}\mathbb{P}l_{\theta^{(0)}}\left(\theta^{(0)}\right)\left(1 - \frac{\gamma c}{8\beta}\right)^t$$

$$\leq \left(1 - \frac{c}{4\beta}\right)\mathbb{P}l_{\theta^{(0)}}\left(\theta^{(0)}\right)\left(1 - \frac{c}{8\beta}\right)^t + \frac{c}{8\beta}\mathbb{P}l_{\theta^{(0)}}\left(\theta^{(0)}\right)\left(1 - \frac{\gamma c}{8\beta}\right)^t$$

$$\leq \mathbb{P}l_{\theta^{(0)}}\left(\theta^{(0)}\right)\left(1 - \frac{c}{8\beta}\right)^{t+1}.$$

Similarly,

$$r\left(\theta^{(t+1)}\right) \geq r^* - c_2\tilde{\mathbb{P}}l_{\theta^{(t+1)}} \geq r^* - c_2\tilde{\mathbb{P}}l_{\theta^{(0)}}\left(1 - \frac{\gamma c}{8\beta}\right)^{t+1}.$$

Consequently, we choose

$$N_t = r\left(\theta^{(t)}\right)^{-1}\left(2\left(16\beta - 2c\right)c^{-1}C_LC_\Theta\log c_\Theta\right)^2\left(\tilde{\mathbb{P}}l_\theta^{(0)}\right)^{-2}\left(1 - \frac{\gamma c}{8\beta}\right)^{-2t},$$

then for the initial value $\theta^{(0)}$ which is sufficiently accurate, we have

$$r\left(\theta^{(t)}\right)^{1/2} \geq r\left(\theta^{(0)}\right)^{1/2} \geq 4N_t^{-1/2}\log N_t,$$

ensuring that our choice of $N_t$ meets the needed condition.

Finally, we define $\zeta = \gamma c/(8\beta - \gamma c)$, then

$$n_t = N_tr\left(\theta^{(t)}\right) = \left(2\left(16\beta - 2c\right)c^{-1}C_LC_\Theta\log c_\Theta\right)^2\left(\tilde{\mathbb{P}}l_\theta^{(0)}\right)^{-2}(1 + \zeta)^{-2t}.$$

By the inequality

$$1 - \sum_{t=0}^{T-1} N_t^{-4} \geq 1 - \sum_{t=0}^{T-1} n_t^{-4}$$

we obtain the results stated in Theorem B.6. □

*Proof of Theorem B.4.* By Assumption B.3, we have

$$\sup_{\theta \in \Theta} \mathbb{E}_{x \sim \mathcal{P}_\theta}[\mathcal{R}(x)] - \mathbb{E}_{x \sim \mathcal{P}_{\theta_{\pi_{\text{const}}}^{(T)}}}[\mathcal{R}(x)] \geq c_3 \mathbb{P} l_{\theta_{\pi_{\text{const}}}^{(T)}} \theta_{\pi_{\text{const}}}^{(T)}$$

$$\geq c_3 \left( \mathbb{P} l_{\theta_{\pi_{\text{const}}}^{(T)}} \theta_{\pi_{\text{const}}}^{(T)} - \mathbb{P}_{n_0} l\left(X; \theta_{\pi_{\text{const}}}^{(T)}\right) \right) + \mathbb{P}_{n_0} l\left(X; \theta_{\pi_{\text{const}}}^{(T)}\right)$$

$$\geq c_3 \frac{1}{n_0} \sum_{i=1}^{n_0} \left( l\left(x_i; \theta_{\pi_{\text{const}}}^{(T)}\right) - \mathbb{E}_{X \sim \mathcal{P}_{l\left(x_i; \theta_{\pi_{\text{const}}}^{(T)}\right)}}\left[ l\left(X; \theta_{\pi_{\text{const}}}^{(T)}\right) \right] \right)$$

$$\geq c_3 n_0^{-1/2} \Phi(0.75) \operatorname{Var}_{X \sim \mathcal{P}_{l\left(x_i; \theta_{\pi_{\text{const}}}^{(T)}\right)}} \left[ l\left(X; \theta_{\pi_{\text{const}}}^{(T)}\right) \right]^2$$

with probability larger than $1 - 0.75 = 0.25$, where we invoke the central limit theorem for the last inequality. □

*Proof of Theorem B.5.* Let $\varepsilon > 0$ be given. We first show that there exists some $t$ satisfying

$$r\left(\theta_\pi^{(t)}\right) \geq r^* - \frac{1}{2}\varepsilon.$$

Following the proof of Theorem B.6, we have

$$\tilde{\mathbb{P}} l_{\theta_\pi^{(t+1)}} - \tilde{\mathbb{P}} l_{\theta_\pi^{(t)}} \leq -\frac{\gamma c}{4\beta} \tilde{\mathbb{P}} l_{\theta_\pi^{(t)}} + \gamma \left(2 - \frac{c}{4\beta}\right) 2 C_L C_\Theta \log c_\Theta n_t^{-1/2}.$$

There exists a sufficiently large $T'$ such that

$$\gamma \left(2 - \frac{c}{4\beta}\right) 2 C_L C_\Theta \log c_\Theta n_T^{-1/2} \leq \min\left\{ \frac{\gamma c}{16 \beta c_2} \varepsilon, \frac{1}{2 c_2} \varepsilon \right\}.$$

For any $t \geq T'$, consider the following two cases:

(a) $\tilde{\mathbb{P}} l_{\theta_\pi^{(t)}} \leq \varepsilon/(2 c_2)$.

By Assumption B.3, we have

$$r\left(\theta_\pi^{(t)}\right) \geq r^* - c_2 \tilde{\mathbb{P}} l_{\theta_\pi^{(t)}} \geq r^* - \frac{1}{2}\varepsilon.$$

(b) $\tilde{\mathbb{P}} l_{\theta_\pi^{(t)}} > \varepsilon/(2 c_2)$.

Since $n_t \to \infty$ as $t \to \infty$, for all $t \geq T'$ we have

$$\gamma \left(2 - \frac{c}{4\beta}\right) 2 C_L C_\Theta \log c_\Theta n_t^{-1/2} \leq \frac{\gamma c}{16 \beta c_2} \varepsilon \leq \frac{\gamma c}{8\beta} \tilde{\mathbb{P}} l_{\theta_\pi^{(t)}}.$$

Therefore,

$$\tilde{\mathbb{P}} l_{\theta_\pi^{(t+1)}} - \tilde{\mathbb{P}} l_{\theta_\pi^{(t)}} \leq -\frac{\gamma c}{8\beta} \tilde{\mathbb{P}} l_{\theta_\pi^{(t)}}.$$

Repeating this argument for $t = T', T' + 1, \cdots$ whenever $\tilde{\mathbb{P}} l_{\theta_\pi^{(t)}} > \varepsilon/(2 c_2)$ shows that there must exist a finite $K$ (so $T = T' + K$) where either

$$r\left(\theta_\pi^{(t)}\right) \geq r^* - c_2 \frac{\gamma c}{4\beta} \tilde{\mathbb{P}} l_{\theta_\pi^{(T)}}$$

$$\geq r^* - c_2 \frac{\gamma c}{4\beta} \tilde{\mathbb{P}} l_{\theta_\pi^{(T')}} \left(1 - \frac{\gamma c}{8\beta}\right)^{T - T'} \geq r^* - \frac{1}{2}\varepsilon,$$

or else $\tilde{\mathbb{P}} l_{\theta_\pi^{(T)}} \leq \varepsilon/(2 c_2)$, in which case (a) implies

$$r\left(\theta_\pi^{(t)}\right) \geq r^* - \frac{1}{2}\varepsilon.$$

Thus, there exists some $T$ such that

$$r\left(\theta_\pi^{(T)}\right) \geq r^* - \frac{1}{2}\varepsilon.$$

Next, we prove that for any $t \geq T$,

$$r\left(\theta_\pi^{(t)}\right) \geq r^* - \varepsilon.$$

For $t \geq T$, we have

$$\sup_{t \geq T} \left[\tilde{\mathbb{P}}l_{\theta_\pi^{(t+1)}} - \tilde{\mathbb{P}}l_{\theta_\pi^{(t)}}\right] \leq \sup_{t \geq T}\left[-\frac{\gamma c}{4\beta}\tilde{\mathbb{P}}l_{\theta_\pi^{(t)}} + \gamma\left(2 - \frac{c}{4\beta}\right)2C_L C_\Theta \log c_\Theta n_t^{-1/2}\right]$$

$$\leq \sup_{t \geq T}\left[\gamma\left(2 - \frac{c}{4\beta}\right)2C_L C_\Theta \log c_\Theta n_t^{-1/2}\right]$$

$$\leq \frac{\varepsilon}{2c_2}.$$

From part (b) above, whenever $\tilde{\mathbb{P}}l_{\theta_\pi^{(t)}} \geq \varepsilon/(2c_2)$, we have

$$\tilde{\mathbb{P}}l_{\theta_\pi^{(t+1)}} - \tilde{\mathbb{P}}l_{\theta_\pi^{(t)}} \leq -\frac{\gamma c}{8\beta}\tilde{\mathbb{P}}l_{\theta_\pi^{(t)}}.$$

Hence,

$$\sup_{t \geq T}\tilde{\mathbb{P}}l_{\theta_\pi^{(t)}} \leq \frac{\varepsilon}{2c_2} + \sup_{t \geq T'}\left[\tilde{\mathbb{P}}l_{\theta_\pi^{(t+1)}} - \tilde{\mathbb{P}}l_{\theta_\pi^{(t)}}\right] \leq \frac{\varepsilon}{c_2}.$$

It follows that for all $t \geq T$,

$$r\left(\theta_\pi^{(t)}\right) \geq r^* - \varepsilon,$$

which completes the proof. $\qquad\square$

*Proof of Theorem B.7.* We prove this theorem by showing that for a given policy $\pi'$, there exists a specified parameter space $\Theta$ and $\Theta^*$, a loss function $l(x; \theta)$, and a reward model $\mathcal{R}(x)$, under Assumptions B.1 to B.3 such that

$$\begin{aligned}
C(\pi', T^*(\pi', \varepsilon)) &= \sum_{t=0}^{T^*(\pi',\varepsilon)-1}(c_g N_t' + c_t n_t') \\
&> C_{\exp}\varepsilon^{-2} - c_{\exp} \\
&\geq C(\pi_{\exp}, T^*(\pi_{\exp}, \varepsilon))
\end{aligned} \tag{9}$$

with probability at least $\left(1 - \sum_{t=0}^{T^*(\pi_{\exp},\varepsilon)-1} n_t^{-4}\right)$.

To begin, we use Theorem B.6 to obtain

$$T^*(\pi_{\exp}, \varepsilon) \leq \log^{-1}\left(1 - \frac{\gamma c}{8\beta}\right)\left(\log \varepsilon - \log c_2 \tilde{\mathbb{P}}l_{\theta^{(0)}}\right) := T_1$$

with probability at least $\left(1 - \sum_{t=0}^{T^*(\pi_{\exp},\varepsilon)-1} n_t^{-4}\right)$. Then the equality

$$\begin{aligned}
C(\pi_{\exp}, T^*(\pi_{\exp}, \varepsilon)) &\leq C(\pi_{\exp}, T_1) \\
&= \sum_{t=0}^{T_1-1}(c_g r_t^{-1} + c_t)\left(2(16\beta - 2c)c^{-1}C_L C_\Theta \log c_\Theta\right)^2\left(\tilde{\mathbb{P}}l_\theta^{(0)}\right)^{-2}\left(1 - \frac{\gamma c}{8\beta}\right)^{-2t} \\
&\leq (c_g r_0^{-1} + c_t)\left(2(16\beta - 2c)c^{-1}C_L C_\Theta \log c_\Theta\right)^2\left(\tilde{\mathbb{P}}l_\theta^{(0)}\right)^{-2}\sum_{t=0}^{T_1-1}\left(1 - \frac{\gamma c}{8\beta}\right)^{-2t} \\
&\leq (c_g r_0^{-1} + c_t)\left(2(16\beta - 2c)c^{-1}C_L C_\Theta \log c_\Theta\right)^2\left(\tilde{\mathbb{P}}l_\theta^{(0)}\right)^{-2}\frac{(1+\zeta)^{2(T_1-1)-1}}{2\zeta + \zeta^2} \\
&\leq C_{\exp}\varepsilon^{-2} - c_{\exp}.
\end{aligned}$$

holds for some positive constants $C_{\exp}$ and $c_{\exp}$ that do not depend on $\varepsilon$. So we have proven the last inequality in Equation (9).

Next, we consider the second inequality in Equation (9). When $\pi'$ is a constant policy, Theorem B.4 implies that for sufficiently small $\varepsilon$, $T^*(\pi', \varepsilon) = \infty$ and the inequality holds vacuously. Otherwise, for a polynomial growth policy $\pi'$ under Assumptions B.1 to B.3, there exist $\Theta$, $\Theta^*$, $l(x; \theta)$, and $\mathcal{R}(x)$ such that for $\theta^{(t)}_{\pi'}$,

$$\tilde{\mathbb{P}}l_{\theta^{(t+1)}_{\pi'}} - \tilde{\mathbb{P}}l_{\theta^{(t)}_{\pi'}} = \min\left\{-\frac{\gamma c}{4\beta}\tilde{\mathbb{P}}l_{\theta^{(t)}_{\pi'}} + \rho\gamma\left(2 - \frac{c}{4\beta}\right)2C_LC_\Theta \log c_\Theta \frac{1}{\sqrt{n'_t}}, 0\right\},$$

for some $0 < \rho \le 1$, and by Assumption B.3,

$$r^* - r\left(\theta^{(t)}_{\pi'}\right) \ge c_3\tilde{\mathbb{P}}l_{\theta^{(t)}_{\pi'}}.$$

Since $\pi'$ is a polynomial growth policy, let $\pi' = (n'_0, n'_1, n'_2, \cdots)$ with $n'_t = n'_0(1+t)^\alpha, \alpha > 0$ by definition. A decrease in expected loss occurs only if

$$\rho\gamma\left(2 - \frac{c}{4\beta}\right)2C_LC_\Theta \log c_\Theta \frac{1}{\sqrt{n'_0(1+t)^\alpha}} \le \frac{\gamma c}{4\beta}\tilde{\mathbb{P}}l_{\theta^{(t)}_{\pi'}}.$$

Besides,

$$\tilde{\mathbb{P}}l_{\theta^{(T^*(\pi', \varepsilon))}_{\pi'}} \le c_3^{-1}\left(r^* - \mathbb{E}_{x \sim \mathcal{P}_{\theta^{(T^*(\pi', \varepsilon))}_{\pi'}}}[\mathcal{R}(x)]\right) \le c_3^{-1}\varepsilon,$$

it follows that

$$\rho\gamma\left(2 - \frac{c}{4\beta}\right)2C_LC_\Theta \log c_\Theta \frac{1}{\sqrt{n'_0(1+T^*(\pi', \varepsilon))^\alpha}} \le \frac{\gamma c}{4\beta c_3}\varepsilon,$$

so

$$T^*(\pi', \varepsilon) \ge \left[\left(\frac{c}{2c_3\rho(8\beta - c)C_LC_\Theta \log c_\Theta}\right)^{-2}n'^{-1}_0\right]^{1/\alpha}\varepsilon^{-2/\alpha} - 1 := T_2.$$

Thus,

$$\sum_{t=0}^{T^*(\pi', \varepsilon)-1}(c_g N'_t + c_t n'_t) \ge \sum_{t=0}^{T_2-1}(c_g N'_t + c_t n'_t)$$

$$\ge \sum_{t=0}^{T_2-1}(c_g + c_t)n'_0(1+t)^\alpha$$

$$\ge (c_g + c_t)n'_0\left(\frac{T_2}{2}\right)^\alpha \frac{T_2}{2} := C_{\text{poly}}\varepsilon^{-2(\alpha+1)/\alpha} - c_{\text{poly}}$$

for some positive constants $C_{\text{poly}}$ and $c_{\text{poly}}$ independent of $\varepsilon$. If $\varepsilon$ is small enough, it follows that

$$\sum_{t=0}^{T^*(\pi', \varepsilon)-1}(c_g N'_t + c_t n'_t) \ge C_{\text{poly}}\varepsilon^{-2(\alpha+1)/\alpha} - c_{\text{poly}} > C_{\exp}\varepsilon^{-2} - c_{\exp},$$

so the second inequality in Equation (9) also holds.

$\square$

## B.5 Extension to Rewards with Random Noise

Our theoretical conclusions directly generalize to cases with noisy rewards. Consider a perturbed reward function $\hat{\mathcal{R}}(x) = \mathcal{R}(x) + \varepsilon$, where $\varepsilon$ is zero-mean noise independent of $\mathcal{R}(x)$. Crucially, the expected reward $r(\theta)$ remains unchanged, preserving all key theoretical results. Although noisy rewards affect the sample selection process, the concentration bounds for selected sample sizes (Appendix B.4) and the expected reward under the selected distribution remain valid asymptotically. This robustness stems from the exponential policy's inherent adaptivity by progressively increasing sample sizes (Theorem 4.3), it naturally compensates for reward noise variance.

## C    Diffusion Probabilistic Models

A diffusion probabilistic model (DPM) progressively adds noise to clean data sampled from the target distribution until the data is fully corrupted. The generative process then employs a noise prediction model to transform noisy samples into progressively cleaner ones auto-regressively. This reverse-time evolution naturally resembles a denoising process, making DPMs suitable for image denoising, especially when the noise modeling is known (Xie et al., 2023).

Formally, a DPM defines a forward stochastic process with conditional probability

$$p_{\tau|0}(\boldsymbol{x}_\tau \mid \boldsymbol{x}_0) = \mathcal{N}(\boldsymbol{x}_\tau; \alpha_\tau \boldsymbol{x}_0, \beta_\tau^2 \boldsymbol{I}), \quad \tau \in [0, 1],$$

where $\boldsymbol{x}_0$ is drawn from the target distribution that the model aims to learn. The parameters $\alpha_\tau$ and $\beta_\tau$ are typically chosen such that $\alpha_0 = \beta_1 = 1$ and $\alpha_1 = \beta_0 = 0$. This process progressively adds noise to $\boldsymbol{x}_0$, resulting in a fully corrupted sample $\boldsymbol{x}_1 \sim \mathcal{N}(\boldsymbol{0}, \boldsymbol{I})$.

To model the reverse process, the time interval $[0, 1]$ is discretized into steps $0 = \tau_0 < \tau_1 < \cdots < \tau_K = 1$. Starting at $\tau_K$, the reverse diffusion process gradually removes noise step by step until reaching $\tau_0$. At each step $k$, the noise added during the forward process is estimated and removed using a trained noise prediction model $\boldsymbol{\varepsilon}_\theta(\cdot, \tau_k)$. This model learns to predict the noise $\boldsymbol{\varepsilon}$ based on the noisy sample $\boldsymbol{x}_{\tau_k}$, enabling the reconstruction of cleaner samples at each step.

The standard training procedure, which is also adapted in our experiments, attempts to solve

$$\min_{\boldsymbol{\theta}} \mathbb{E}_{\boldsymbol{x} \sim \mathscr{D}} \mathbb{E}_{\substack{\boldsymbol{\varepsilon} \sim \mathcal{N}(\boldsymbol{0}, \boldsymbol{I}) \\ \tau \in \mathcal{U}(0, s)}} \left\| \boldsymbol{\varepsilon}_{\boldsymbol{\theta}}(\alpha_\tau \boldsymbol{x} + \beta_\tau \boldsymbol{\varepsilon}, \tau) - \boldsymbol{\varepsilon} \right\|^2, \tag{10}$$

where $\mathscr{D}$ denotes the distribution of training data.

## D    Algorithm for Realistic Experiments

We follow the algorithm in Algorithm 1. Empirically, in the two realistic settings (image denoising and math reasoning), our iterative learning policy satisfies that $n_t$ is an integer multiple of the batch size. Specifically,

- For the constant scheme,
$$n_t = \lfloor n \rfloor B;$$

- For the linear scheme,
$$n_t = \lfloor n(t+1) \rfloor B;$$

- For the exponential scheme,
$$n_t = \lfloor n(1+u)^t \rfloor B,$$

where $n$ and $u$ serve as hyperparameters, and $B$ is the generation batch size.

## E    Implementation Details

We provide the code for all the experiments in the GitHub repository: `https://github.com/zylipku/spend-wisely`.

### E.1    Toy Example

In this subsection, we extend the theoretical intuition from Theorem 3.1 to high-dimensional settings and then derive the expected reward of each iteration, as well as the corresponding optimal expected reward.

We begin with the following theorem, which shows that in a high-dimensional Gaussian setting, the same exponential growth strategy emerges as in the one-dimensional case.

**Theorem E.1.** *Under the MLE update, suppose that the generator is a d-dimensional Gaussian distribution $\mathcal{N}(\boldsymbol{\theta}, \sigma^2 \boldsymbol{I}_d)$, the reward model is an exponential function $\mathcal{R}(\boldsymbol{x}) = \exp\left(-\|\boldsymbol{x}\|^2/(2\kappa^2)\right)$, and we start from*

$$\left\| \boldsymbol{\theta}^{(0)} \right\| \le \left(1 + \sigma^2/\kappa^2\right)^T \left(d\left(\sigma^2 + \kappa^2\right)\right)^{1/2}.$$

*Considering the optimization problem*

$$\max_{\pi} \quad \mathbb{E}_{\boldsymbol{\theta}^{(T)}} \left[ \mathbb{E}_{\boldsymbol{x} \sim \mathcal{P}_{\boldsymbol{\theta}^{(T)}}} \mathcal{R}(\boldsymbol{x}) \right] \quad s.t. \quad \sum_{k=0}^{T-1} n_t \leq C,$$

*then the optimal policy satisfies*

$$n_t \propto \left( 1 + \frac{\sigma^2}{\kappa^2} \right)^t.$$

*Proof.* Since each component of $\boldsymbol{\theta}^{(t)}$ is i.i.d. and its distribution conditioned on the initial parameter $\theta^{(0)}$ and the policy $\pi$ is given by Lemma A.1, it follows that

$$p\left( \boldsymbol{\theta}^{(T)} \mid \boldsymbol{\theta}^{(0)}, \pi \right) = \mathcal{N} \left( \frac{\boldsymbol{\theta}^{(0)}}{(1 + \sigma^2/\kappa^2)^T}, \sigma^2 \sum_{t=0}^{T-1} \frac{1}{n_t (1 + \sigma^2/\kappa^2)^{2(T-t)-1}} \boldsymbol{I}_d \right)$$

for a specific policy $\pi : \{n_t\}_t$. By the definition of the reward model, $\mathcal{R}(\boldsymbol{x})$ factors as

$$\mathcal{R}(\boldsymbol{x}) = \prod_{i=1}^{d} \mathcal{R}(x_i), \quad \boldsymbol{x} = (x_1, x_2, \cdots, x_d).$$

Repeating the argument in the proof of Theorem 3.1 shows that

$$\mathbb{E}_{\boldsymbol{\theta}^{(T)}} \mathbb{E}_{\boldsymbol{x}} \mathcal{R}(\boldsymbol{x}) = \mathbb{E}_{\boldsymbol{\theta}^{(T)}} \mathbb{E}_{\boldsymbol{x}} \prod_i \mathcal{R}(x_i)$$

$$= \prod_i \mathbb{E}_{\boldsymbol{\theta}^{(T)}} \mathbb{E}_{\boldsymbol{x}} \mathcal{R}(x_i)$$

$$= \left( \frac{\kappa}{\sqrt{\sigma^2 + \kappa^2 + \sigma_T^2}} \right)^d \exp \left( -\sum_{i=1}^{d} \frac{\left( \theta_i^{(0)} \right)^2}{2(\sigma^2 + \kappa^2 + \sigma_T^2)(1 + \sigma^2/\kappa^2)^{2T}} \right)$$

$$= \left( \frac{\kappa}{\sqrt{\sigma^2 + \kappa^2 + \sigma_T^2}} \right)^d \exp \left( -\frac{\left\| \boldsymbol{\theta}^{(0)} \right\|^2}{2(\sigma^2 + \kappa^2 + \sigma_T^2)(1 + \sigma^2/\kappa^2)^{2T}} \right).$$

With assumption $\|\boldsymbol{\mu}_T\| \leq \left( d(\sigma^2 + \kappa^2) \right)^{1/2}$, the right side increases as $\left( \sigma^2 + \kappa^2 + \sigma_T^2 \right)^{-1/2}$ increases. Hence, figuring with the proof of Theorem 3.1 in Appendix A, one obtains

$$n_t \propto \left( 1 + \frac{\sigma^2}{\kappa^2} \right)^{t-T}.$$

Reindexing or shifting $t$ completes the proof. $\square$

Theorem E.1 therefore shows that the optimal iterative policy in the high-dimensional case mirrors the one-dimensional scenario. Next, we derive the expected reward in the high-dimensional case. Specifically, if $\boldsymbol{x} \sim \mathcal{N}(\boldsymbol{\theta}, \sigma^2 \boldsymbol{I}_d)$, then

$$\mathbb{E}_{\boldsymbol{x} \sim \mathcal{N}(\boldsymbol{\theta}, \sigma^2 \boldsymbol{I}_d)}[\mathcal{R}(\boldsymbol{x})] = \int \frac{1}{(2\pi\sigma^2)^{d/2}} \exp \left( -\frac{\|\boldsymbol{x}\|^2}{2\kappa^2} - \frac{\|\boldsymbol{x} - \boldsymbol{\theta}\|^2}{2\sigma^2} \right) d\boldsymbol{x}$$

$$= \frac{1}{(1 + \sigma^2/\kappa^2)^{d/2}} \cdot \exp \left( -\frac{\|\boldsymbol{\theta}\|^2}{2(\sigma^2 + \kappa^2)} \right).$$

Clearly, the maximum of this expression is attained at $\boldsymbol{\theta} = \boldsymbol{0}$, and the optimal expected reward is $(1 + \sigma^2/\kappa^2)^{-d/2}$.

### E.2 Synthetic Experiment with Gaussian Distributions

**Setup.** We begin with the synthetic setting introduced in our warm-up (Section 3.2), which involves optimizing a generative distribution parameterized as a Gaussian to maximize the expected reward. Specifically, we focus on a two-dimensional space using MLE updates at each iteration. The generator is set to $\mathcal{N}(\boldsymbol{\theta}, \boldsymbol{I}_2)$, with fixed variance. The reward model is defined as an exponential function $\mathcal{R}(\boldsymbol{x}) = \exp\left(-\|\boldsymbol{x}\|^2/(2\kappa^2)\right)$, favoring data concentrated near the origin. $\kappa$ controls the flatness of the reward.

All theoretical results from the warm-up extend naturally to the high-dimensional case, as detailed in Appendix E.1. We now validate that with equal computational budgets and iterations, the exponential scheme outperforms the linear scheme, which outperforms the constant scheme.

**Implementation.** We consider two settings with $\kappa^2 \in \{2, 4\}$. Based on theoretical analysis, the exponential scheme is set as $n_t = \lfloor n_0 \cdot (1 + 1/\kappa^2)^t \rfloor$, where $n_0 = 10$. For a fixed total budget and number of iterations, the constant scheme is then

$$
n_t = \left\lfloor T^{-1} \sum_{k=0}^{T-1} n_0 \cdot (1 + 1/\kappa^2)^k \right\rfloor,
$$

while the linear scheme is

$$
n_t = \left\lfloor \frac{2(t+1)}{T(T-1)} \sum_{k=0}^{T-1} n_0 (1 + 1/\kappa^2)^k \right\rfloor.
$$

The initial parameter is set to $\boldsymbol{\theta}^{(0)} = (1, 1)^\top$, and the optimal solution for the reward function is $\boldsymbol{\theta}^* = (0, 0)^\top$. The optimal reward value is then

$$
\mathbb{E}_{\boldsymbol{x} \sim \mathcal{N}(\boldsymbol{0}, \boldsymbol{I})}[\mathcal{R}(\boldsymbol{x})] = \left(1 + 1/\kappa^2\right)^{-1}.
$$

We evaluate how fast the three schemes approach this optimal reward value.

### E.3 Image Denoising

The pre-trained model we choose is a public checkpoint that can be accessed via `https://huggingface.co/google/ddpm-cifar10-32`. For all denoising experiments, we fix batch size $B = 640$ and learning rate $5 \times 10^{-5}$.

At each iteration, synthetic samples are generated on the fly, scored by a reward model

$$
\mathcal{R}(\hat{\boldsymbol{x}}, \boldsymbol{x}) = \frac{\text{PSNR}(\hat{\boldsymbol{x}}, \boldsymbol{x}) - r_{\min}}{r_{\max} - r_{\min}},
$$

and accumulated into a chunk until it reaches size $n_t$. The full training procedure is given in Algorithm 2. Each configuration requires around 100 GPU-hours on an NVIDIA A800 cluster. Note that our setup is much more computationally intensive than the usual training of a generative model in that it takes $s$ times of network inferences to generate synthetic data for network training.

To compare increasing-schedule policies (particularly exponential growth), we report the computational costs in terms of floating-point operations (FLOPs), which scale linearly with the number of forward passes. For diffusion models, generating one sample uses $s$ denoising steps (i.e., $s$ forward passes), and each optimization step introduces an additional forward pass.

### E.4 Math Reasoning

We summarize the iterative math-reasoning algorithm in Algorithm 3. Each configuration requires approximately 400 GPU-hours on an NVIDIA A800 cluster.

Hyperparameters are as follows: one epoch per iteration on the selected data; a constant learning rate of $10^{-7}$; batch size $B = 256$; and roughly 1,000 total generator update steps. During generation we use temperature 0.3 for diversity, and temperature 0 at evaluation for accuracy. The maximum generation length is 512. Prompts follow the few-shot setting of the `GSM8k` dataset in OpenCompass (Contributors, 2023). Our implementation builds on the public OpenRLHF framework (Hu et al., 2024).

---

**Algorithm 2** Iterative learning for image denoising

---

1: **Input:** a pre-trained generative model $f(\cdot; \boldsymbol{\theta}^{(0)})$, a policy $\pi : \{n_t\}_t$ with a terminal time $T$, a
   reward model $\mathcal{R}$, and a dataset $\mathcal{D} = \{\boldsymbol{x}_i\}_i$.
2: **Output:** a generative model $f(\cdot; \boldsymbol{\theta}^{(T)})$ fine-tuned with synthetic data.

3: **for** $t \leftarrow 0$ to $T - 1$ **do**
4:     Initialize the set of selected synthetic data $D_t = \varnothing$.

5:     **while** $|D_t| < n_t$ **do**
6:         Sample a mini-batch $\{\boldsymbol{x}_i\}_i \subseteq \mathcal{D}$ randomly.
7:         Create noisy $\boldsymbol{y}_i \sim p_{s|0}(\cdot \mid \boldsymbol{x}_i)$ for each $\boldsymbol{x}_i$.
8:         Generate synthetic $\hat{\boldsymbol{x}}_i$ as a denoised version for each $\boldsymbol{y}_i$ based on the generative model
           $f(\cdot; \boldsymbol{\theta}^{(t)})$;
9:         For each $\hat{\boldsymbol{x}}_i$, add it into $D_t$ with probability $\mathcal{R}(\hat{\boldsymbol{x}}_i, \boldsymbol{x}_i)$ clipped within the interval $[0, 1]$.
10:     **end while**

11:     Truncate $D_t$ so that it contains exactly $n_t$ samples.

12:     **for** each batch $S = \{\hat{\boldsymbol{x}}_i\}_i$ in $D_t$ **do**
13:         Take a single optimization step with the loss

$$L\left(\boldsymbol{\theta}^{(t)}\right) = |S|^{-1} \sum_{\boldsymbol{x} \in S} \|\boldsymbol{\varepsilon}_{\boldsymbol{\theta}^{(t)}}(\alpha_{\tau_{\boldsymbol{x}}} \boldsymbol{x} + \beta_{\tau_{\boldsymbol{x}}} \boldsymbol{\varepsilon}_{\boldsymbol{x}}, \tau_{\boldsymbol{x}}) - \boldsymbol{\varepsilon}_{\boldsymbol{x}}\|^2$$

        to update $\boldsymbol{\theta}^{(t)}$, where $\boldsymbol{\varepsilon}_{\boldsymbol{x}} \sim \mathcal{N}(\boldsymbol{0}, \boldsymbol{I})$ and $\tau_{\boldsymbol{x}} \sim \mathcal{U}(0, s)$.
14:     **end for**

15:     $\boldsymbol{\theta}^{(t+1)} \leftarrow \boldsymbol{\theta}^{(t)}$.
16: **end for**

---

# F   Supplementary Empirical Results

## F.1   Toy Example

Additional results for the toy example, obtained under various parameter settings, are shown in
Figure 5. In all cases, the behavior is consistent with the conclusion in Section 3.2.

## F.2   Ablation on Exponential Growth Schemes in Image Denoising

We conduct an ablation study for exponential growth policies to understand the impact of the $n_t$ on
the performance of the iterative learning framework, as measured by PSNR in Figure 6. The results
reveal that a smaller $n_t$ during the early stages of training leads to a rapid increase in PSNR and
earlier convergence. In contrast, a larger $n_t$ results in a slower initial increase in PSNR but achieves a
higher final PSNR when trained long enough. These findings underscore the trade-off between rapid
early improvements and the potential for superior overall performance with prolonged training.

## F.3   Ablation on Exponential Growth Schemes in Math Reasoning

We further explore the effects of varying the parameter $n_t$ within exponential growth policies on the
performance of mathematical reasoning tasks. As shown in Figure 7, configurations with higher $n_t$
demonstrate the potential for superior performance, improving the model's reasoning capabilities and
ability to solve mathematical problems. However, these configurations also exhibit some instability,
characterized by abrupt changes in accuracy as $n_t$ increases. This highlights the need to find a balance
between achieving rapid learning and mitigating the risk of unstable training dynamics. Based on
these findings, we adopt $n_t = 10 \cdot 2^t \cdot B$ in the main text.

## F.4   Additional Empirical Results in Math Reasoning

In the main paper, we use a temperature of 0.3 for the math reasoning experiment shown in Figure 4.
Here, we present additional empirical results with a temperature of 0.7 in Figure 8. The general

**Algorithm 3** Iterative learning on math reasoning

---

1: **Input:** a pre-trained LLM $f(\cdot; \theta^{(0)})$, a policy $\pi : \{n_t\}_t$ with a terminal time $T$, and a dataset $\mathcal{D} = \{(q_i, a_i)\}_i$.
2: **Output:** $f(\cdot; \theta^{(T)})$ fine-tuned with synthetic data.

3: **for** $t \leftarrow 0$ to $T - 1$ **do**
4:      Initialize the set of selected synthetic data $D_t = \varnothing$.

     // Generate answers and select out correct ones.
5:      **while** $|D_t| < n_t$ **do**
6:          Sample a mini-batch $\mathcal{B} = \{(q_j, a_j)\}_j \subseteq \mathcal{D}$.
7:          Generate answers $\{\hat{a}_j\}$ based on $f(\cdot \mid \{q_j\}; \theta^{(t)})$ with few-shot examples and CoT strategy.
8:          Add $(q_j, a_j)$ into $D_t$ if $\hat{a}_j = a_j$ for each $j$.
9:      **end while**

10:      $D_t \leftarrow$ a subset of $D_t$ containing exactly $n_t$ data.

     // Update $\theta^{(t)}$ to $\theta^{(t+1)}$ by the selected dataset $D_t$ with auto-regressive loss via Adam optimizer.
11:      **for** each $\mathcal{B}$ as a mini-batch of $D_t$ **do**
12:          Take a single optimization step with the auto-regressive loss

$$L(\theta) = \frac{1}{|\mathcal{B}|} \sum_{(\hat{a},q) \in \mathcal{B}} \log p(\hat{a} \mid q; f(\cdot, \theta))$$

         to update $\theta^{(t)}$.
13:      **end for**
14:      $\theta^{(t+1)} \leftarrow \theta^{(t)}$.
15: **end for**

---

Table 3: Additional empirical results of mixed policies in the math reasoning task. For the mixed policies, we use a linear policy initially and switching to an exponential policy when the linear phase ceased to grow. The results show that mixed policies outperform linear policies and are comparable to the exponential policies.

| POLICY | SYMBOLIC | P1 | P2 |
|--------|----------|-----|-----|
| LINEAR | 64.24 | 40.84 | 19.17 |
| MIXED | **65.98** | 45.56 | 20.16 |
| EXPONENTIAL | 65.66 | **47.26** | **20.65** |

trends across different curves and policy classes remain consistent when comparing these two figures. This ablation study further demonstrates the superiority of the proposed approach.

### F.5 Mixed Policies

We also explore mixed policies empirically. Specifically, we use a linear policy initially, switching to an exponential policy when the linear phase ceases to yield growth. We present the results in Table 3, where we directly combine the linear and exponential schemes. The results show that mixed policies do not outperform the pure exponential policy. While this approach could be seen as a more general strategy, the increased degrees of freedom in parameter tuning make hyperparameter selection considerably more challenging. We leave further exploration of this direction to future work.

## G  Discussion

### G.1  Impact Statement

This research advances the field of machine learning by addressing the strategic allocation of computational resources in the post-training phase using synthetic data. We introduce a novel theoretical

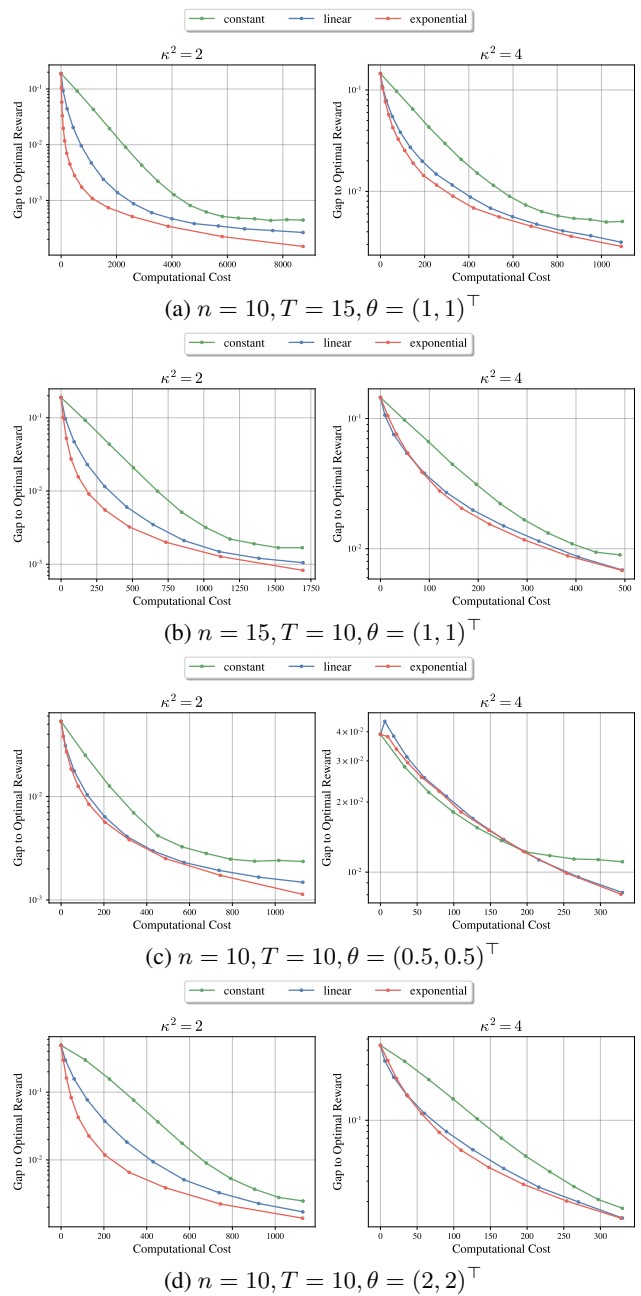

Figure 5: Additional experimental results of the toy example with diverse parameters.

framework to optimize the distribution of a fixed budget across multiple training iterations, an approach that demonstrates significant improvement in model performance. Our findings challenge conventional constant allocation strategies, revealing that dynamically adjusted policies, particularly those with exponential growth, not only converge more stably but also do so at a reduced computational cost. This work has broad implications for the development of AI models, especially in domains where the generation of synthetic data is critical for enhancing model capabilities without the extensive need for expensive or hard-to-obtain real-world data. The practical applications of this research extend to fields such as autonomous driving, medical image analysis, and any other area where AI must perform with high stability and efficiency under constraints of limited labeled data. This study thus provides a foundation for more sustainable AI development practices that prioritize both performance and computational efficiency.

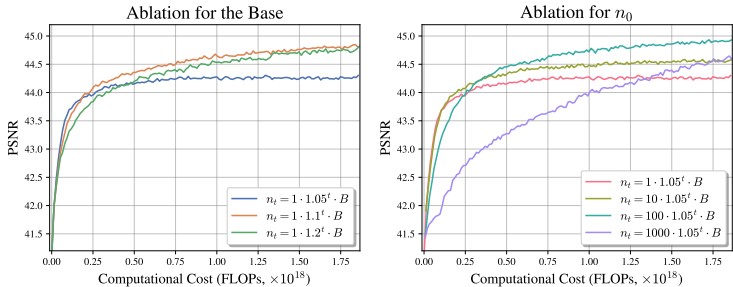

Figure 6: Ablation study for the image denoising task. We show the performance of exponential scheme under the $s = 10$ setting with different parameters.

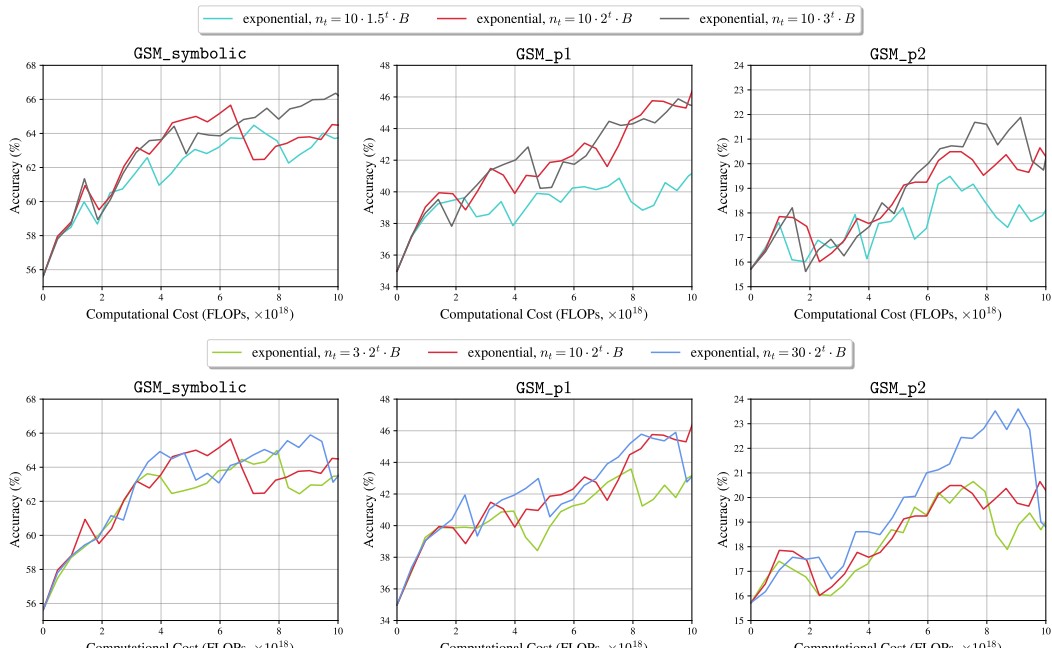

Figure 7: Ablation study for the math reasoning task. We show the performance of exponential scheme with different parameters.

## G.2 Connection with Model Collapse

Prior work has introduced verification (Feng et al., 2024; El Firdoussi et al., 2025) and correction (Gillman et al., 2024) mechanisms for synthetic data to mitigate model collapse. They have theoretically derived one-step conditions under which collapse is avoided or triggered. Our methodology crucially relies on the reward model-based data verification process, using reward signals to filter data to prevent model collapse.

Empirically, we observe that increasing policies are generally more resistant to collapse. For constant policies, even with reward-based curation, the model's performance initially improves but then degrades over successive iterations. Increasing polices yield smoother, generally upward trajectories, though some decline under very extensive finetuning. These observations extend beyond the scope of the analyses in (Feng et al., 2024; El Firdoussi et al., 2025) and suggest the need for a deeper theoretical analysis. We will add this discussion on model collapse to the paper and leave it for future work.

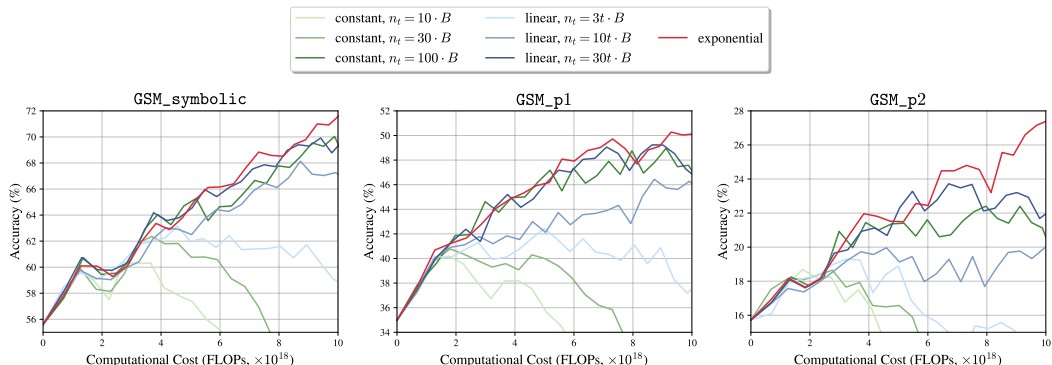

Figure 8: Additional empirical results of math reasoning where the temperature is 0.7. Other hyperparameters remain the same them in Figure 4, i.e., the training batch size is still set to $B = 256$ and the exponential policy is still $n_t = 10 \cdot 2^t \cdot B$.

## G.3 Limitation

In this paper, we focus on achieving efficient SFT improvement within an iterative learning framework. Recent trends (Guo et al., 2025) incorporate reinforcement learning into the entire post-training phase, while SFT remains an important component for enabling the model to exhibit certain behaviors and skills necessary for successful RL training later (Gandhi et al., 2025). We aim to extend the theory to develop an efficient training scheme for RLHF and RL from verifiable rewards.

We also examine learning with access to a reward model with (Appendix B.5) or without independent random noises. An empirical reward, such as using an LLM as a judge, could produce biased results and lead to reward hacking. We leave the exploration of biased rewards to future work.

We conduct two experiments to validate our theoretical results. However, incorporating the SFT scheme design into production scale is beyond our resources at academic institutions. We present theoretical results and experimental findings across modalities that support the generalizability of our results.

