# OpenReview forum: "Spend Wisely: Maximizing Post-Training Gains in Iterative Synthetic Data Bootstrapping"
_NeurIPS.cc/2025/Conference — NeurIPS 2025 spotlight_

### Official Review · Reviewer_9EDy · 2025-06-30

**Clarity:** 4
**Significance:** 3
**Originality:** 3
**Rating:** 5
**Confidence:** 2

**Summary:**

This paper investigates iterative synthetic data bootstrapping for post-training of generative foundation models such as LLM, utilizing synthetic data and existing reward models. In this setting, synthetic data generated from the generative model is input into the reward model at each iteration, and the decision to adopt it as training data is made based on its reward score. Subsequently, the model is updated through supervised fine-tuning (SFT) using the synthetic training data. This bootstrapping method has already been theoretically analyzed, and it is known to converge optimally under the assumption of infinite samples. This paper theoretically analyzes the behavior of iterative synthetic data bootstrapping in a more realistic problem setting constrained by a fixed budget. Specifically, it addresses the research question of how many synthetic samples should be generated at each iteration. To this end, the paper conducts (i) a theoretical analysis of a Gaussian distribution toy model and (ii) a theoretical analysis under realistic settings with mild assumptions, demonstrating that exponentially increasing the number of synthetic samples per iteration is optimal in terms of convergence to optimal performance. Experiments provide observational results supporting this theoretical finding through diffusion models and LLM post-training.

**Questions:**

Could you answer or comment on the concerns in the weakness sections?

**Ethical Concerns:**

["NO or VERY MINOR ethics concerns only"]

**Final Justification:**

I appreciate the authors' responses. The explanations in the rebuttal have largely resolved my concerns regarding hyperparameters. I recommend accepting this paper.

**Limitations:**

The paper discusses the application of paradigms other than SFT and the application to production scale as limitations in Section B. These discussions are adequate.

**Paper Formatting Concerns:**

Nothing to report.

**Quality:**

4

**Strengths And Weaknesses:**

### Strengths
+ **S1.** The paper is well structured and clearly describes the research questions and contributions. Post-training is an important phase in operating foundation models such as LLM, and reducing its cost is a topic of interest. The theoretical results of this paper are not limited to toy models but are also demonstrated in practical settings, indicating a wide range of applications.
+ **S2.** The empirical experiments to validate the theory are appropriately designed.
+ **S3.** There is potential for further development toward more practical settings. While this study focuses on SFT, expanding this discussion could advance theoretical analysis of post-training using RLHF or GRPO in the future.
### Weaknesses
- **W1.** The practicality of the experimental results is somewhat weak, as discussed in Section B. In particular, the paper would have had a greater impact if there had been experiments suggesting the extent of the effect on general post-training of LLM rather than on specialized tasks. However, I understand that the main contribution of the paper is theoretical analysis and its verification, and please note that this comment does not request these experiments for the rebuttal.
- **W2.** In order to properly implement an exponential growth policy, hyperparameter tuning and a validation dataset are necessary. Investigating the extent to which this affects performance in realistic settings with a small validation set would further highlight the strengths of this paper.

---

> ### Author Rebuttal · Authors · 2025-07-31
>
> We sincerely thank the reviewer for their thorough and encouraging assessment. We greatly appreciate the recognition of our paper's clear structure, well-defined research questions, and significant contributions to the important field of post-training cost reduction for foundation models. We are also grateful for the acknowledgment that our theoretical results are not limited to toy models but are demonstrated in practical settings, indicating a wide range of applications.
>
> > [W1] The practicality of the experimental results is somewhat weak, as discussed in Section B. In particular, the paper would have had a greater impact if there had been experiments suggesting the extent of the effect on general post-training of LLM rather than on specialized tasks. However, I understand that the main contribution of the paper is theoretical analysis and its verification, and please note that this comment does not request these experiments for the rebuttal.
>
> We thanks for the reviewer's understanding. Our aim is to conduct rigorous, scientifically sound comparisons at scales our academic resources allow, articulate the resulting insights clearly for the community, and leave it for the engineering teams at large companies to integrate them into frontier-scale post-training.
>
> > [W2] In order to properly implement an exponential growth policy, hyperparameter tuning and a validation dataset are necessary. Investigating the extent to which this affects performance in realistic settings with a small validation set would further highlight the strengths of this paper.
>
> We appreciate the reviewer’s thoughtful comment. We fully agree that in practice, implementing a good exponential growth policy requires choosing an appropriate increasing rate, and it is hard to find the optimal value a priori.
>
> That said, our results still recommend an exponential schedule (Theorem 4.4). Both constant and polynomial policies also require careful tuning of parameters (e.g., number of samples for the initial iteration, growth rate), yet the exponential policy carries a provable compute-efficiency advantage.
>
> In our ablation studies, we observe that exponential policies is reasonably robust to variations in hyperparameters in Figure 7 and 8: increasing the base from 2 to 3 shifts the peak performance by less than 1 %. This suggests that while tuning is necessary, the exponential strategy does not require fine-grained search to yield strong performance in practice.
>
> We thank the reviewer once again for the time and effort. Please let us know if we have addressed all the questions.

---

> > ### Comment · Reviewer_9EDy · 2025-08-04
> >
> > Thank you for clarifying. The rebuttal fully addresses my concerns. This work has a strong impact on future research on post-training with synthetic datasets, so I raised my confidence score by 1.

---

> > > ### Author Response · Authors · 2025-08-06
> > >
> > > We sincerely thank the reviewer for the encouraging follow-up and the recognition of the potential impact of our work. We are glad to see that our rebuttal has addressed your concerns, and please do not hesitate to reach out if further clarifications are needed.

---

### Official Review · Reviewer_hQEz · 2025-07-01

**Clarity:** 3
**Significance:** 3
**Originality:** 3
**Rating:** 5
**Confidence:** 3

**Summary:**

This paper addresses the budget allocation problem for iterative synthetic data bootstrapping. Iterative synthetic data bootstrapping is a training process where models generate synthetic data, filter the data using reward models, and fine-tune the model iteratively to maximize expected rewards. The authors develop a theoretical framework to analyze optimal allocation of computational resources across iterations to maximize model performance. Their main contribution is proving that constant allocation policies fail to converge with high probability, while exponential growth policies achieve optimal performance with exponential convergence rates. The theoretical analysis is validated through experiments on Gaussian toy problems, image denoising with diffusion models on MNIST, and mathematical reasoning with LLMs on synthetic arithmetic datasets.

**Questions:**

* Did the authors explore mixed policies—for example, using a linear policy initially and switching to an exponential policy if the target performance is not achieved? This could offer a more flexible strategy in practice.

* Have the authors evaluated the performance of the exponential policy in scenarios where the assumptions underlying the theoretical analysis are violated? It would be useful to understand how robust the method is in such cases.

* The caption for Figure 1 appears to be merged with the image. Could the authors correct this formatting issue in the next revision?

**Ethical Concerns:**

["NO or VERY MINOR ethics concerns only"]

**Final Justification:**

This paper tackles the budget‐allocation problem in iterative synthetic‐data bootstrapping—a critical component of tuning models to optimize a given reward function. In light of its solid theoretical and practical contributions, I vote to accept and encourage the authors to include results on more complex scenarios—such as reward‐based optimization of text‐to‐image models—in the final version.

**Limitations:**

yes

**Paper Formatting Concerns:**

The paper does not have any formatting concerns.

**Quality:**

3

**Strengths And Weaknesses:**

### **Strengths**

* The paper presents a clear and well-motivated problem formulation for an important task. It appears to be the first systematic evaluation of budget allocation in iterative synthetic data bootstrapping. However, I cannot fully assess the novelty of the contribution, as this is not my primary area of expertise.

* Given that training on synthetic data is a key component of modern generative modeling pipelines, the proposed method has the potential for substantial practical impact.

* The experiments span both vision and language domains, demonstrating the generality and versatility of the approach.

* The paper is well-written and easy to follow.

* The theoretical analysis is thorough and well-structured, offering insights into both the benefits of the exponential policy and the convergence issues with constant policies.

### **Weaknesses**

* The primary weakness is the limited scope of experimental evaluation. The vision experiments are restricted to MNIST, and the language experiments focus on basic arithmetic tasks. Additionally, the use of oracle reward models reduces the realism and practical relevance of the setup. While this limitation alone is not a reason for rejection, the paper would be significantly strengthened by experiments on more complex and realistic tasks.

* It is not clear to me how model collapse is prevented when fine-tuning on synthetic datasets using this method. The experiments suggest that additional fine-tuning continues to improve performance, but it would be interesting to see more analysis on when, and under what conditions, the model begins to degrade due to excessive fine-tuning on synthetic data.

---

> ### Author Rebuttal · Authors · 2025-07-31
>
> We sincerely thank the reviewer for their comprehensive and thoughtful comments. We highly appreciate the recognition of our paper's clear and well-motivated problem formulation for a timely task, and its systematic evaluation of budget allocation in iterative synthetic data bootstrapping. We are also grateful for the acknowledgment of our demonstration on the generality and versatility of our approach.
>
> > [W1] The primary weakness is the limited scope of experimental evaluation. The vision experiments are restricted to MNIST, and the language experiments focus on basic arithmetic tasks. Additionally, the use of oracle reward models reduces the realism and practical relevance of the setup. While this limitation alone is not a reason for rejection, the paper would be significantly strengthened by experiments on more complex and realistic tasks.
>
> We thanks the reviewer for acknowleding our experiments "span both vision and language domains, demonstrating the generality and versatility of the approach".
>
> Regarding the choice of datasets. Our guiding principle was to select datasets that allow a rigorous test of the theory while remaining tractable under academic-scale compute: MNIST is one of the image benchmarks on which diffusion models have not been extensively pre-trained, making it a clean test bed for iterative bootstrapping effects. GSM-Symbolic is a de-contaminated math dataset that remains challenging for 8B instruct models. These settings let us demonstrate our method across two modalities (images, text) and model families (diffusion, LLM) in a controlled environment.
>
> Regarding the oracle reward. We work in regimes where a reliable reward (oracle or natural) is available. Our contribution is to show how, given such a reward model, one can maximize bootstrapping gains. Learning a high-quality reward proxy when no natural signal exists is, in our mind, an orthogonal problem.
>
> > [W2] It is not clear to me how model collapse is prevented when fine-tuning on synthetic datasets using this method. The experiments suggest that additional fine-tuning continues to improve performance, but it would be interesting to see more analysis on when, and under what conditions, the model begins to degrade due to excessive fine-tuning on synthetic data.
>
> Prior work has introduced verification [1,2] and correction [3] mechanisms for synthetic data to mitigate model collapse. They have theoretically derived one-step conditions under which collapse is avoided or triggered. Our methodology crucially relies on the reward model-based data verification process, using reward signals to filter data to prevent model collapse.
>
> Empirically, we observe that increasing policies are generally more resistant to collapse. For constant policies, even with reward-based curation, the model's performance initially improves but then degrades over successive iterations. Increasing polices yield smoother, generally upward trajectories, though some decline under very extensive finetuning. These observations extend beyond the scope of the analyses in [1,2] and suggest the need for a deeper theoretical analysis. We will add this discussion on model collapse to the paper and leave it for future work.
>
> [1] Feng, Y., Dohmatob, E., Yang, P., Charton, F., and Kempe, J. Beyond model collapse: Scaling up with synthesized data requires reinforcement. arXiv preprint arXiv:2406.07515, 2024.
>
> [2] Firdoussi, Aymane El, et al. "Maximizing the Potential of Synthetic Data: Insights from Random Matrix Theory." arXiv preprint arXiv:2410.08942 (2024).
>
> [3] Gillman, Nate, et al. "Self-correcting self-consuming loops for generative model training." arXiv preprint arXiv:2402.07087 (2024).
>
> > [Q1] Did the authors explore mixed policies—for example, using a linear policy initially and switching to an exponential policy if the target performance is not achieved? This could offer a more flexible strategy in practice.
>
> Thanks for this suggestion. Mixed policies, such as initiating with a linear strategy and adaptively switching to an exponential one, are indeed flexible practial solution. We will try the following hyperparameter tuning strategy: we begin with a modest growth rate, then gradually increase the growth rate when performance (evaluated on a small validation dataset) stops growing. Given the computational demands of this experiment, we will include the results in the final version.
>
> > [Q2] Have the authors evaluated the performance of the exponential policy in scenarios where the assumptions underlying the theoretical analysis are violated? It would be useful to understand how robust the method is in such cases.
>
> Thanks for this question. We appreciate the reviewer’s interest in understanding the robustness of our method beyond the theoretical assumptions. Our current theoretical framework is built on a relatively mild set of assumptions.
>
> The core dependence of our theoretical results are on Assumption B.1. This condition is standard in prior learning-theory analyses, e.g., on the application of covering numbers [1,2,3], and on the continuity and convexity of the loss function [4,5,6].
>
> For general settings, it is inherently difficult to verify whether the assumptions are satisfied, or to analytically characterize their violation. This challenge is not unique to our work, but reflects a broader limitation in theoretical research, where it is often necessary to make simplifying assumptions in order to derive nontrivial insights. We believe this is a reasonable tradeoff, especially since our assumptions are not tailored to any specific experimental design.
>
> All of our empirical results—from both vision and language domains—align with the theoretical insights, further supporting their practical robustness.
>
> [1] Zhou, Ding-Xuan. "The covering number in learning theory." Journal of Complexity 18.3 (2002): 739-767.
>
> [2] Suzuki, Taiji, Hiroshi Abe, and Tomoaki Nishimura. "Compression based bound for non-compressed network: unified generalization error analysis of large compressible deep neural network." International Conference on Learning Representations. 2020.
>
> [3] Jin, Ming, et al. "On solution functions of optimization: Universal approximation and covering number bounds." Proceedings of the AAAI Conference on Artificial Intelligence. Vol. 37. No. 7. 2023.
>
> [4] Bassily, Raef, et al. "Private stochastic convex optimization with optimal rates." Advances in neural information processing systems 32 (2019).
>
> [5] Zou, Difan, et al. "Gradient descent optimizes over-parameterized deep ReLU networks." Machine learning 109 (2020): 467-492.
>
> [6] Yang, Long, et al. "Policy optimization with stochastic mirror descent." Proceedings of the AAAI Conference on Artificial Intelligence. Vol. 36. No. 8. 2022.
>
> > [Q3] The caption for Figure 1 appears to be merged with the image. Could the authors correct this formatting issue in the next revision?
>
> Thanks for pointing it out. We will correct it in the next version.
>
> We thank the reviewer once again for the time and effort. Please let us know if we have addressed all the questions.

---

> > ### Comment · Reviewer_hQEz · 2025-08-04
> >
> > I would like to thank the authors for their response. I still believe it would be interesting to apply the tuning strategy to a more complex vision model (such as reward-based tuning of text-to-image generative models), as such applications could further strengthen the pipeline. That said, I also view the paper as a major theoretical contribution, and I understand that such practical experiments may be beyond the current scope. My other questions have been addressed in the rebuttal, and I will increase my score accordingly.

---

> > > ### Author Response · Authors · 2025-08-07
> > >
> > > We sincerely thank the reviewer for acknowledging the theoretical contribution of our work and raising the score accordingly. We also agree that exploring under more complex scenarios could further demonstrate the practical utility and scalability, and we are actively considering this line of work for future research.

---

### Official Review · Reviewer_mn1i · 2025-07-02

**Clarity:** 2
**Significance:** 2
**Originality:** 3
**Rating:** 4
**Confidence:** 2

**Summary:**

The paper explores in iterative supervised fine-tuning where the generated samples from the current model are filtered with certain reward functions and then used for the next iteration of training, actually needs a linear or exponential number of samples as more iterations are performed given a fixed overall budget. Empirical studies also support the theoretical findings in the math reasoning and umage denoising tasks.

**Questions:**

1. In theorem 4.2 and 4.3, it implies the error with go down as we increase T which is natual, but doesn't it imply the number of samples/total budget will grow for polynomial growth or exponetial growth policies? How can they use the same budget as the constant policy? Or is $n_0$ from growing policies much smaller than the constant policy?
2. Since getting a large number of synthetic datapoints is easy, have you tried not limiting the budget and just using more examples in grwoth policies and does that give even better performance?
3. I am not aware of similar arguments in RL that the fixed number of samples per iter is suboptimal. Could you comment on the intuition about the differences?

**Ethical Concerns:**

["NO or VERY MINOR ethics concerns only"]

**Final Justification:**

Thank you for the rebuttal. It mostly solved the concern and I will keep my score.

**Limitations:**

Yes

**Paper Formatting Concerns:**

In Figure 1 the title and the figure is overlapped.

**Quality:**

2

**Strengths And Weaknesses:**

Strengths:
1. The paper has very interesting findings both theoretically and empirically. It basically says that the model will need more samples for later stages in the iterations when doing self-bootstrapped training via theoretical insights.
2. It seems novel since I am not aware of other works doing this analysis.
3. The writing is generally clear and easy to understand.


Weaknesses:
1. The experimental results have no error bars.
2. The range of the tasks in the experiments is somehow limited.
2. In theorem 4.2 and 4.3 the role of the constraint of the overall budget is confusing. See Q1.

---

> ### Author Rebuttal · Authors · 2025-07-31
>
> We sincerely thank the reviewer for their comprehensive and highly encouraging comments. We are particularly grateful for the recognition of our paper's very interesting and novel findings, both theoretically and empirically. We appreciate the acknowledgment of our core insight that models benefit from an increasing number of samples in later stages of iterative self-bootstrapped training, supported by rigorous theoretical analysis.
>
> > [W1] The experimental results have no error bars.
>
> We agree that including error bars would further strengthen the analysis. However, providing error bars for iterative bootstrapping experiments, especially for generating data and training 8B models, consume more compute than we can access in an academic lab.
>
> Within these constraints, we chose to spend our budget on breadth rather than repetition: we benchmarked the methods across a diverse set of tasks and carried out an extensive ablation study in Figure 6, 7, and 8. We believe this strategy still offers compelling evidence of the effectiveness and generality of our approach.
>
> > [W2] The range of the tasks in the experiments is somehow limited.
>
> We would like to respectfully clarify that our empirical validation was deliberately designed to cover diverse settings, thereby providing robust support for our theoretical findings:
>
> 1. **Theoretical validation**: The Gaussian toy experiment provides controlled verification of Theorem 3.1 under multiple parameter configurations.
>
> 2. **Image denoising**: The image denoising task evaluates two different noise levels ($s=0.01$ vs $0.02$) with comprehensive ablation studies on exponential policy parameters (Fig. 6).
>
> 3. **Math reasoning**: Crucially, our LLM experiments employ *three progressively challenging math datasets* (GSM-symbolic, GSM-p1, GSM-p2) specifically designed to test generalization across difficulty levels. We further validate robustness through ablations:
>    - Multiple policy hyperparameters (Fig. 7)
>    - Different generation temperatures (T=0.3 vs 0.7 in Fig. 4 vs Fig. 8)
>    - Full-weight fine-tuning of Llama-3-8B (non-trivial computational scale)
>
> We hope this combination of cross-domain validation (vision and language) and the wide coverage of ablations provide strong empirical evidence for our conclusions.
>
> > [W3 & Q1] In theorem 4.2 and 4.3 the role of the constraint of the overall budget is confusing. ... In theorem 4.2 and 4.3, it implies the error with go down as we increase T which is natual, but doesn't it imply the number of samples/total budget will grow for polynomial growth or exponetial growth policies? How can they use the same budget as the constant policy? Or is $n_0$ from growing policies much smaller than the constant policy?
>
> Thank you for raising this point. We would like to clarify how the overall budget constraint comes into our theoretical development.
>
> **Theorems 4.1–4.3.** These three results analyze the convergence of the respective policies (constant, polynomial, and exponential) as the number of iterations (T) grows. Theorem 4.1 shows that constant policies generally fail to converge with high probability. Theorems 4.2 and 4.3 demonstrate the convergence of polynomial and exponential policies, respectively. At this stage, the analysis tracks error decay without imposing any limit on total compute budget.
>
> **Introducing the Budget.** Only after obtaining the unconstrained convergence results, we introduce the budget constraint from line 205 and present the theory in **Theorem 4.4**. Equation (2) defines the minimum number of iterations each policy requires to achieve a target reward. Theorem 4.4 compares the computational cost of different policies and shows that, under a fixed budget, the exponential schedule reaches the target reward most efficiently, outperforming both the constant and polynomial schedules.
>
> We will improve the presentation of this section, emphasising that the budget constraint enters only after the initial convergence analysis.
>
> > [Q2] Since getting a large number of synthetic datapoints is easy, have you tried not limiting the budget and just using more examples in growth policies and does that give even better performance?
>
> Building on the previous point, Theorems 4.1–4.3 show that, in the absence of any compute constraint, growth policies converge better.
>
> In reality, "getting a large number of synthetic datapoints" online in iterative bootstrapping is not easy: at every iteration, the current model needs to generate new synthetic data. For both language models (using next token prediction) and diffusion models, the cost of generation can, in fact, significantly exceed the cost of model training:
>
> * For Diffusion Models, generating each sample necessitates a sequence of many denoising steps, each involving a forward pass through the model.
>
> * For LLMs, the serial nature of inference (multiple forward propagations are required to generate each token) results in an overall time cost that may instead be slower than training [1,2].
>
> Therefore, we consider the realistic setup where there is a budget constraint. No matter how large the budget, Theorem 4.4 shows that the exponential schedule yields the best performance for a fixed total compute.
>
> [1] Kim, Kyoungmin, et al. "Optimizing LLM Inference for Database Systems: Cost-Aware Scheduling for Concurrent Requests." arxiv preprint arxiv:2411.07447 (2024).
>
> [2] Shashank and Neal. "Mastering LLM Techniques: Inference Optimization" Technical Blog of Generative AI
>
> > [Q3] I am not aware of similar arguments in RL that the fixed number of samples per iter is suboptimal. Could you comment on the intuition about the differences?
>
> This is a really good question. Please allow us to restate our insights: we demonstrate that constant policies fail to converge to optimality with high probability, while increasing policies converge. This crucial difference arises directly from **the noise introduced by finite samples in the gradient estimation step**, and the constant sample size brings noises in gradient estimation whose variance does not decrease as the number of iterations increases.
>
> A similar issue also exists in RL. Many theoretical RL works often assume access to a "ground truth" / noiseless policy gradient at each step, without considering the direct impact of finite sample noise [1,2]. Empirical RL often employs techniques such as replay buffers to enlarge the sample size with offline data to help stabilize the optimization. However, this approach also has its limitations and requires extensive tuning.
>
> In our self-improvement scenario, we find that the noise contributed by finite samples is particularly significant. An increasing policy can progressively reduce this noise and stabilize the learning process, leading to convergence.
>
> [1] Agarwal, Alekh, et al. "On the theory of policy gradient methods: Optimality, approximation, and distribution shift." Journal of Machine Learning Research 22.98 (2021): 1-76.
>
> [2] Xiao, Lin. "On the convergence rates of policy gradient methods." Journal of Machine Learning Research 23.282 (2022): 1-36.
>
> We thank the reviewer once again for the time and effort. Please let us know if we have addressed all the questions.

---

> > ### Comment · Reviewer_mn1i · 2025-08-07
> >
> > Thank you for the rebuttal. It mostly solved the concern and I will keep my score.

---

> > > ### Author Response · Authors · 2025-08-07
> > >
> > > We are glad to see that the rebuttal resolved most of your concerns, and we sincerely thank you for supporting our work.

---

### Official Review · Reviewer_5fr2 · 2025-07-02

**Clarity:** 4
**Significance:** 2
**Originality:** 3
**Rating:** 5
**Confidence:** 4

**Summary:**

Multiple studies have shown that generative models can be retrained on their own (carefully curated) outputs in order to enhance their performances, which can be done iteratively multiple times. This paper tackles the question of how to split your training budget across the retraining iterations in order to maximize the performances of the generative model. The paper proposes experiments on toy data, image generation and coding tasks.

**Questions:**

- I have a question on the theoretical setup and the experimental protocol:
    - The differences between the policy only lies in the number of sample at each step? If this is the case, could the variation  in performance of each method only be du to the number of samples with large reward in the training set?
- Do the empirical and theoretical results suggest adding samples with large rewards as soon as possible (over the retraining iterations) in the training set?

**Ethical Concerns:**

["NO or VERY MINOR ethics concerns only"]

**Final Justification:**

I think this is a good paper, hence I keep my score: 5

**Limitations:**

Yes

**Quality:**

3

**Strengths And Weaknesses:**

Strengths:
- The study tackles a very timely problem: how to spend you retraining budget?
- The paper is clearly written

Weaknesses:
- As mentioned by the authors themselves, experimental evaluation can be seen as limited in this setup: except for the code generation experiment, for which a natural meaningful reward exists. IMO, this weakness is not critical

---

> ### Author Rebuttal · Authors · 2025-07-31
>
> We sincerely thank the reviewer for their thoughtful evaluation and for highlighting the key strengths of our work. We are especially grateful for the recognition that our study tackles a timely and important problem concerning the optimal allocation of training budgets in iterative synthetic data bootstrapping. We also appreciate the reviewer’s positive remarks on the clarity and quality of our manuscript.
>
> > [W1] As mentioned by the authors themselves, experimental evaluation can be seen as limited in this setup: except for the code generation experiment, for which a natural meaningful reward exists. IMO, this weakness is not critical
>
> We agree that our study is limited to scenarios in which a natural or oracle reward signal is directly available. Our contribution is deliberately scoped: given such a reward model, we investigate how to maximize bootstrapping gains. By contrast, learning a high-quality reward proxy when no natural signal exists is, in our mind, an orthogonal problem. Recent work continues to push reward modelling forward on that front by training judges and leveraging rubrics.
>
> > [Q1&Q2] The differences between the policy only lies in the number of sample at each step? If this is the case, could the variation in performance of each method only be du to the number of samples with large reward in the training set? Do the empirical and theoretical results suggest adding samples with large rewards as soon as possible (over the retraining iterations) in the training set?
>
> Indeed, the primary difference among our proposed strategies lies in the number of synthetic samples curated at each iteration. Our theory shows that constant policy will suffer from the noise introduced by the fixed-size sample set, preventing reliable convergence. Therefore, we propose increasing policy, which generates more and more samples across iteration to control the noise down in order to finally converge stably.
>
> The resulting performance differences are not due to the number of samples with high rewards, and our empirical and theoretical results do not suggest simply "adding samples with large rewards as soon as possible in the training set". To clarify, we cannot **actively** or **directly** inject high-reward samples into the training set in our setup. As highlighted in the box at line 103, the curation pipeline is identical across all settings: the model under training generates candidates, and the same reward model filters them to curate good examples. This mechanism is fixed; the sole variable is the total number of training samples per iteration, not the number of large-reward ones. To be optimal in budget, the sample size should grow progressively with each iteration rather than start out high "as soon as possible".
>
> We thank the reviewer once again for the time and effort. Please let us know if we have addressed all the questions.

---

> > ### Comment · Reviewer_5fr2 · 2025-08-01
> >
> > I thank the authors for the clarifications.
> > I think this is a good paper and I will keep my score

---

> > > ### Author Response · Authors · 2025-08-06
> > >
> > > We sincerely thank the reviewer for continuing support for our work.

---

### Decision · Program_Chairs · 2025-09-17

**Decision:**

Accept (spotlight)

**Comment:**

In post training of generative models (LLM or Diffusion Models), synthetic data training with Verifier selection is a key paradigm. This paper answers the question that what should be the scheduling policy of number of generated synthetic data points given a total budget. Theoretical analysis are made, from simple Gaussian data to practical reward models, and concluded exponential policy on the scheduling works the best. The theories are also supported by empirical results on both image generation and LLM math reasonings. All reviewers uniformly agreed that this paper has good contribution and should be accepted.